# LaViDa: A Large Diffusion Language Model for Multimodal Understanding

**Shufan Li**[1*], **Konstantinos Kallidromitis**[2*], **Hritik Bansal**[1*], **Akash Gokul**[4*], **Yusuke Kato**[2]
**Kazuki Kozuka**[2], **Jason Kuen**[3], **Zhe Lin**[3], **Kai-Wei Chang**[1], **Aditya Grover**[1]
[1]UCLA [2]Panasonic AI Research [3]Adobe Research [4]Salesforce Research
* Equal Contribution

## Abstract

Modern Vision-Language Models (VLMs) can solve a wide range of tasks requiring visual reasoning. In real-world scenarios, desirable properties for VLMs include fast inference and controllable generation (e.g., constraining outputs to adhere to a desired format). However, existing autoregressive (AR) VLMs like LLaVA struggle in these aspects. Discrete diffusion models (DMs) offer a promising alternative, enabling parallel decoding for faster inference and bidirectional context for controllable generation through text-infilling. While effective in language-only settings, DMs' potential for multimodal tasks is underexplored. We introduce LaViDa, a family of VLMs built on DMs. We build LaViDa by equipping DMs with a vision encoder and jointly fine-tune the combined parts for multimodal instruction following. To address challenges encountered, LaViDa incorporates novel techniques such as complementary masking for effective training, prefix KV cache for efficient inference, and timestep shifting for high-quality sampling. Experiments show that LaViDa achieves competitive or superior performance to AR VLMs on multi-modal benchmarks such as MMMU, while offering unique advantages of DMs, including flexible speed-quality tradeoff, controllability, and bidirectional reasoning. On COCO captioning, LaViDa surpasses Open-LLaVa-Next-Llama3-8B by +4.1 CIDEr with $1.92\times$ speedup. On bidirectional tasks, it achieves +59% improvement on Constrained Poem Completion. These results demonstrate LaViDa as a strong alternative to AR VLMs. Code and models is available at https://github.com/jacklishufan/LaViDa

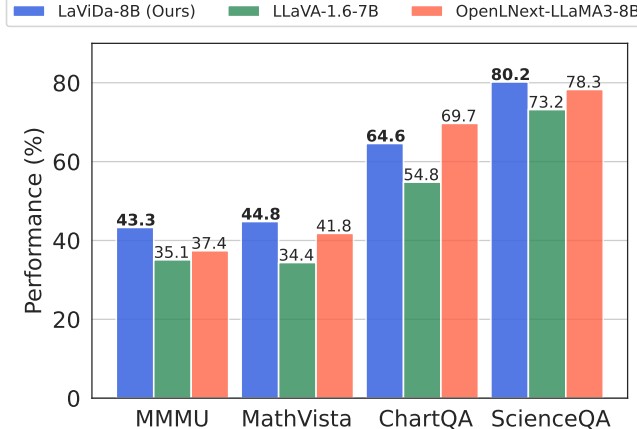

Figure 1: **We propose LaViDa,** the first family of diffusion-based discrete VLM models. LaViDa models achieve competitive performance against AR baselines (LLaVa-1.6, Open-LLaVa-Next) across multiple visual understanding tasks including MMMU (world knowledge), MathVista (reasoning), ChartQA (OCR), and ScienceQA (science).

39th Conference on Neural Information Processing Systems (NeurIPS 2025).

# 1 Introduction

Vision-Language Models (VLMs) have shown remarkable utility across diverse domains, from end-user applications like virtual assistants [37], to research tasks such as scientific image captioning and document understanding [46, 16, 5, 1, 48]. In industrial settings, VLMs support automated product tagging, content moderation, and quality control in manufacturing [69, 2]. Their ability to jointly process visual and textual information makes them indispensable for practical applications and cutting-edge research. Currently, nearly all popular VLMs—such as Qwen-VL [5, 75], Intern-VL [17, 85], and GPT-4 [58] are built on top of large language models (LLMs) that generate text in an autoregressive (AR) manner; that is, they produce tokens one by one in a left-to-right sequence.

While these models have demonstrated strong performance on many tasks, they suffer from several key limitations. First, their sequential generation process is inherently hard to parallelize, resulting in slow inference speed [8]. More critically, their left-to-right generation makes it difficult to handle tasks that benefit from bidirectional context or structural constraints—such as text infilling [22]. For example, generating a poem where each line starts with a specific syllable, or extracting structured information from an image in a predefined JSON format, often requires the model to fill in or coordinate content across the sequence. Autoregressive models still struggle to consistently satisfy such constraints, even with carefully crafted prompts and examples.

Recently, discrete diffusion models (DMs) have emerged as a promising alternative to AR LLMs. Most notably, LLaDA [57] and Dream [79] achieved comparable results to AR LLMs across diverse language tasks. Unlike AR LLMs, DMs treat text generation as a diffusion process over discrete tokens. A forward process gradually corrupts a sequence of discrete text tokens into a sequence of mask tokens. At inference, we start with a sequence of mask tokens and gradually transform them into a sequence of meaningful text tokens through a learned reverse process.

Compared to AR LLMs, diffusion models offer several theoretical advantages that directly address the limitations of autoregressive generation. While AR LLMs have a fixed throughput—generating one token at a time—DMs allow flexible control over the speed-quality trade-off by adjusting the number of diffusion steps [79, 64, 49, 57]. Moreover, their ability to model bidirectional context makes them well-suited for tasks like text infilling, enabling more effective constrained generation and structured output formatting—capabilities especially valuable in vision-language settings where outputs may need to follow specific schemas.

In this work, we propose LaViDa (**La**rge **Vi**sion-Language **D**iffusion Model with M**a**sking), the first family of VLMs based on diffusion. LaViDa enables pretrained DMs to perceive visual inputs by integrating vision features into the diffusion backbone via a vision encoder—analogous to how LLaVA [46, 44] augments large language models (LLMs) with visual inputs. Specifically, we adopt a two-stage training pipeline with a diffusion objective: pretraining followed by supervised fine-tuning.

Adapting DMs for vision-language tasks presents several practical challenges. First, standard DM training is data-inefficient: the model learns only from the subset of corrupted tokens at each timestep. For example, in a question-answering task where the target answer is ''The answer is dog'', the corruption process may mask ''The [mask] is dog'', leaving the key token ''dog'' unmasked and thus excluded from the loss. This is especially problematic for multimodal tasks, where answer tokens often carry crucial semantic content grounded in the image [61]. To address this, we introduce a complementary masking scheme that ensures every token in the output sequence contributes to learning, improving data efficiency.

Second, inference algorithms used by existing DMs are slow in practice due to the lack of KV cache support—an inherent limitation of bidirectional context modeling [3]. This leads to repeated recomputation over the full prompt at every decoding step. While tolerable for short-text settings, it becomes a significant bottleneck in vision-language tasks, where multimodal prompts may include hundreds of visual tokens. To address this, we propose Prefix-DLM decoding, a simple yet effective technique that enables caching of multimodal prompts (i.e., image and question tokens), significantly accelerating inference.

Lastly, DMs offer a unique advantage over autoregressive models: the ability to trade off speed and quality by varying the number of diffusion steps. However, the widely used linear masking schedule—which unmasks a fixed number of tokens per step—performs poorly at low step counts. Motivated by text-to-image diffusion models like SD3 [21], we adopt a timestep shifting strategy that

adaptively adjusts how many tokens are unmasked per step. This leads to better sample quality under aggressive step reduction, allowing faster generation without large degradation in quality.

We conducted extensive evaluations of LaViDa across a wide range of vision-language tasks. Results show that LaViDa achieves competitive performance on most benchmarks, including MMMU [80], MathVista [51], ChartQA [53] and ScienceQA [52], when compared with AR VLMs like LLaVa-1.6-7B [45, 43] and Open-LLaVa-Next-Llama3-8B [15]. We highlight these results in Figure 1. On constrained generation tasks, LaViDa greatly outperforms AR baselines (+59% on Poem Completion). It also supports flexible speed-quality tradeoff, achieving higher quality on COCO image captioning (+ 4.1 CIDEr) and a $1.92\times$ speedup [42]. In summary, our contributions are:

- We introduce LaViDa, the first family of VLMs based on diffusion models. Our models achieve competitive performance across a wide range of tasks compared to AR VLMs, while offering the unique benefits of DMs.

- We introduce several novel training and inference techniques for DMs, including complementary masking, Prefix-DLM, and timestep shifting that improve the training efficiency, inference speed, and sample quality of LaViDa.

- We conduct a systematic study of various design choices for adapting DMs to vision-language tasks (e.g. image resolution), offering insights for future work in this direction.

## 2    Background and Related Works

### 2.1    Vision-Language Models

Vision-Language Models (VLM) extend the capability of Large Language Models to visual understanding tasks [36, 46, 75, 5, 9, 85, 58]. The common recipe to build a VLM is to start with a strong large language model and combine it with a vision encoder [46, 75]. These VLMs typically undergo multiple stages of training, which can be generally categorized into pretraining and finetuning stages. The pretraining data usually consists of text-image pairs for vision-language alignment, while the finetuning data consists of a wide range of instruction-following tasks. There are several dedicated lines of work focusing on different components of this overarching framework, such as the design of vision encoders [26, 77] and training techniques [41, 72]. To this date, most vision-language models such as LLaVa [46, 36] and Qwen-VL [5, 75] series employ an autoregressive training objective.

### 2.2    Diffusion Language Models

Diffusion models first emerged as a powerful alternative to GANs for generating continuous data such as images [62, 59, 21]. Early explorations in diffusion language models directly built continuous diffusion models for latent text embeddings [39, 50], with limited success. More recently, discrete diffusion models [4, 64, 49] have proven to be better candidates for language modeling, achieving performance comparable to AR models while offering unique advantages, such as speed-quality tradeoffs and controllability. Most notably, LLaDa-8B and Dream-8B [57, 79] demonstrated that DLMs can achieve competitive performance against AR LLMs at scale.

Formally, given a text sequence of $L$ tokens $X_0 = [X_0^1, X_0^2...X_0^L]$, the forward process $q(X_t|X_s)$ gradually converts it to a sequence full of mask tokens "[M]", denoted by $X_1 = [X_1^1, X_1^2...X_1^L]$, through the continuous time-interval $[0, 1]$, with $1 \geq t \geq s \geq 0$. A neural network $p_\theta$ is used to model the reverse process $p(X_s|X_t)$. The diffusion language modeling objective can be defined as:

$$\mathcal{L}_{\text{DLM}} = -\mathbb{E}_{t,X_0,X_t}\left[\frac{1}{t}\log p_\theta(X_0|X_t)\right] \tag{1}$$

where $\log p_\theta(X_0|X_t)$ is assumed to be factorized into $\prod_{i=1}^{L} p_\theta(X_0^i|X_t)$. At each training step, we sample $t$ uniformly from the interval $[0, 1]$ and sample $X_0$ from some data distribution $\mathcal{D}$. $X_t$ is then sampled from the forward process $q(X_t|X_0)$. The loss is only computed over the masked tokens in $X_t$, since $p_\theta(X_0^i|X_t)$ has a closed form representation that does not depend on $\theta$ when $X_t^i \neq [M]$. We offer additional background on the details of these formulations in Appendix A.1. We also incorporate addition backgrounds on other relevant literature, such as masked generative models [10, 11] and multimodal diffusion models [68, 38], in Appendix C.

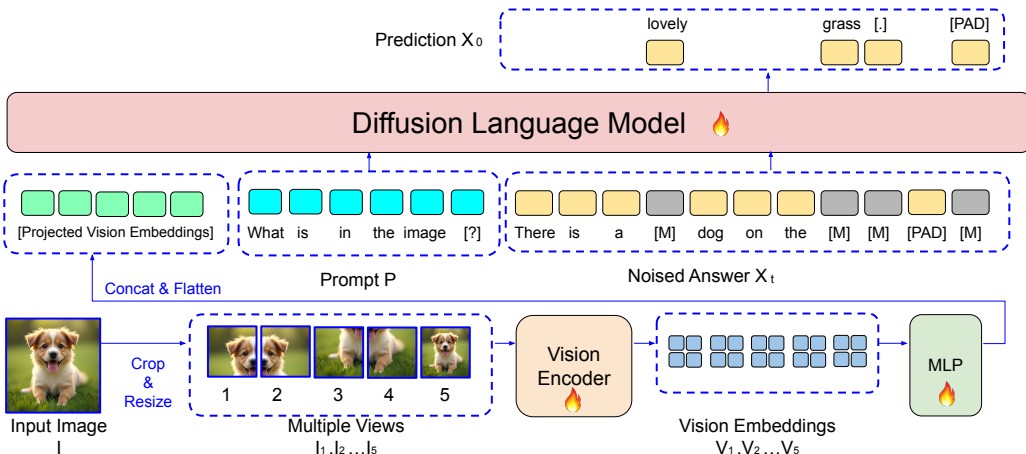

Figure 2: **Overall design of LaViDa.** LaViDa's architecture consists of a vision encoder, a diffusion language model, and an MLP vision projector. The bottom half of the figure illustrates the image encoding process, while the top half depicts the diffusion language modeling process. These two pipelines are described in detail in Sec. 3.1.

## 3 Method

### 3.1 Model Architecture

LaViDa's model architecture follows a similar design to common AR VLMs like LLaVa [36]. It consists of a vision encoder and a diffusion language model. These two parts are connected by a MLP projection network. The overall design is illustrated in Figure 2.

**Vision Encoder**. Given an input image $I$ and text prompt $P$, we first resize the image to $768^2$ and divide it into four non-overlapping views of $384^2$, denoted $I_{1:4}$. We also resize the original image to $384^2$ to obtain a fifth view, $I_5$, following the design of prior works [36, 45]. These five views are independently encoded by the vision encoder (SigLIP-400M [82]), each producing $27^2$ embeddings, denoted $V_{1:5}$. In total, this yields 3645 embeddings per image. To reduce sequence length for efficient training, we apply $2 \times 2$ average pooling on each view, reducing embeddings to $14^2$ per view, or 980 total. The embeddings of five views are then flattened and concatenated into a 1D sequence before being processed by the projection network to obtain the final visual context of the diffusion language model. This process mirrors the vision encoding process in AR LLMs [26] and is illustrated in the bottom part of Figure 2.

**Diffusion Language Model**. The diffusion language model is a multi-layer Transformer [71] whose architecture resembles that of LLMs. The only major difference is that its attention mask is non-causal, and it uses the diffusion language modeling objective described in Section 2.2 instead of the next-token prediction used in AR models. The input to the diffusion language model consists of the projected vision embeddings, the prompt $P$, and partially masked response $X_t$. The outputs of the last transformer block are passed through a final linear layer to obtain token-wise logits $p_\theta(X_0^i|I, P, X_t)$ for the unmasked response $X_0$. In our experiments, we explored LLaDA-8B (default) and Dream-7B as our diffusion language model. This process is illustrated in the upper half of Figure 2.

### 3.2 Training Algorithms

Our training objective is based on the diffusion language modeling objective described in Section 2.2. Each training sample consists of an image $I$, text prompt $P$ and clean text answer $X_0$ from the training data. For multi-round conversation, we sample one round as the "answer" and treat the history as "prompt". We first sample a timestep $t \in [0, 1]$ and a partially masked answer $X_t$ using the forward process described in Section 2.2. LaViDa then implements the conditioned reverse process $p_\theta(X_0|I, P, X_t)$. The canonical diffusion vision-language modeling objective is formulated as:

$$\mathcal{L}_{\text{D-VLM}} = -\mathbb{E}_{t,I,P,X_0,X_t}\left[\frac{1}{t}\log p_\theta(X_0|I, P, X_t)\right] \tag{2}$$

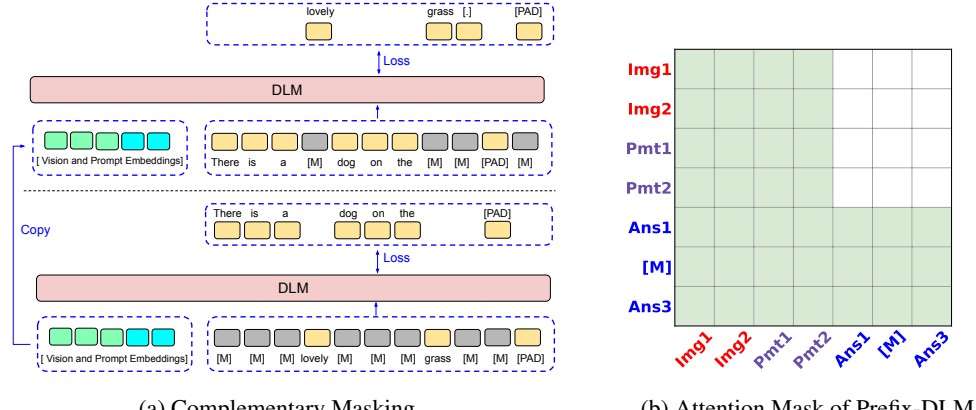

(a) Complementary Masking

(b) Attention Mask of Prefix-DLM

Figure 3: **Technical Details of LaViDa.** (a) We propose Complementary Masking to ensure loss is calculated over all tokens in the data for training efficiency. (b) We propose Prefix-DLM attention mask that enables KV caching. We visualize the attention mask of image tokens (Img1-2), prompt tokens (Pmt1-2), and text and mask tokens in the noise answer $X_t$ (Ans1, [M], Ans3). Rows represent queries, while columns are keys. Colored squares indicate that queries and keys can interact.

where $p_\theta(X_0|I, P, X_t)$ factorizes into $\prod_{i=1}^{L} p_\theta(X_0^i|I, P, X_t)$ following the formulation in Section 2.2. Notably, the loss is only computed over mask tokens where $X_t^i = [M]$, because $p_\theta(X_0^i|I, P, X_t)$ is not dependent on $\theta$ when $X_t^i \neq [M]$.

**Complementary Masking.** Prior diffusion language models (e.g., LLaDa, Dream) apply a stochastic estimator to Equation 2, masking tokens independently across samples in a batch. However, for vision-language tasks, this leads to inefficiencies: (1) only ~50% of tokens contribute to the loss on average, and (2) critical answer tokens—often short and sparse in vision tasks like VQA—may not be masked, resulting in misaligned gradients for the vision encoder. For example, in `"The answer is dog."` the key token `"dog"` might be unmasked in $X_t$ and thus ignored in loss computation. To address this, we introduce complementary masking: for each sample, we generate two masked versions $X_t$ and $X_t^C$ with disjoint corrupted spans (e.g., one masks `"The [M] [M] dog ."`, the other `"[M] answer is [M] [M]"`), ensuring all tokens are eventually used in training and improving sample efficiency and gradient flow. When computing the loss over $X_t$ and $X_t^C$, we copy the encoded vision embeddings to further boost training efficiency. This process is illustrated in Figure 3a.

### 3.3 Inference Algorithms

At inference time, we first create a sequence of $L$ mask tokens as $X_1$, where $L$ is the response generation length. Then we gradually unmask them through $K$ discrete timestamps $t_1..t_K$, where $t_1 = 1$ and $t_K = 0$, until we reach a clean, mask-free sequence $X_0$. At each timestamp $t_i$, we sample a fully unmasked sequence through $p_\theta(X_0|X_{t_i})$ and re-mask $L \times t_{i+1}$ tokens to obtain $X_{t_{i+1}}$. Both $L$ and $K$ are hyperparameters for inference. Additionally, we define $\frac{K}{L}$ as "fraction of the number of functional evaluations (NFE)" to measure sample efficiency. For example, when NFE = 100%, the diffusion model generates one token per forward pass; at NFE = 50%, it generates an average of two tokens per forward pass. Overall, the inference process of LaViDa is similar to prior DMs such as LLaDa, with two key exceptions:

**Prefix-DLM.** While DLMs theoretically offer superior speed–quality tradeoffs at inference, they are often slower than AR models in practice because they cannot leverage KV caching [57]. This issue is particularly evident for multimodal prompts containing many visual tokens. To avoid recomputing keys and values for the visual embeddings and text prompts, we propose a novel Prefix-DLM scheme inspired by the autoregressive prefix-LM. Prefix-DLM adopts a specialized attention mask in which visual and prompt tokens can only attend to other visual and prompt tokens, while answer tokens can attend to all tokens. Figure 3b illustrates this setup. With this design, we can cache the keys and values of the visual and prompt tokens. Empirically, this leads to a speedup of up to 3.9× on COCO captioning tasks. Further details are provided in Section 4.5.

**Schedule Shift.** Diffusion language models (DLMs) allow trading speed for quality via the number of discretization steps $K$. Prior models like LLaDa and Dream use a linear schedule, unmasking $\frac{L}{K}$ tokens uniformly over $t \in [0, 1]$. However, we find this leads to performance degradation at low sampling steps. Inspired by SD3 [21], we adopt a shifting schedule:

$$t'_i = s_\alpha(t_i) = \frac{\alpha t_i}{1 + (\alpha - 1)t_i} \tag{3}$$

Here, $s_\alpha(t)$ is a monotonic map with $t_0 = t'_0 = 0$, $t_K = t'_K = 1$. When $\alpha < 1$ (we use $\alpha = \frac{1}{3}$), the schedule is convex—leading to more tokens being unmasked earlier. We found that this setup outperforms alternatives. Notably, this conclusion differs from that of continuous diffusion models like SD3 and previous masked diffusion models for image generation [11], which showed concave schedules ($\alpha > 1$) are more preferable. We ensure at least one token is unmasked per step. Further details are provided in Section 4 and Appendix A.2.

## 4 Experiments

### 4.1 Setup

At a high level, LaViDa employs a two-stage training process. In the pretraining phase (stage-1), only the projector is updated to align the visual embeddings with the latent space of the DLM. In the finetuning phase (stage-2), we jointly train all components end-to-end for instruction-following. Additionally, we further finetune the stage-2 model for additional steps and obtain two specialized models for reasoning and text-infilling tasks (LaViDa-Reason and LaViDa-FIM). We provide more details of these specialized models in Section 4.3 and 4.4. We use 558K image-text pairs as our stage-1 data, and 1M visual instruction-following examples as our stage-2 data. Further details on the dataset and training setup are provided in Appendix B.

We evaluate LaViDa on a wide range of vision-language tasks. Unless otherwise stated, we report results obtained using the stage-2 model with LLaDa-8B as the language backbone. We use `lmms-eval` package [83] for evaluation and set the sequence length $L$ to be the maximum generation length used for evaluating AR models. We set $K = L$, or NFE=100% by default. Results under differet NFE are explored in Section 4.5 and Appendix B.3 and B.4.

### 4.2 Main Results

Table 1 reports the results of LaViDa using LLaDA-8B (LaViDa-L) and Dream-7B (LaViDa-D) as the language backbones on vision-understanding tasks. We compare with several open-source, open-data models with similar data sizes and parameter counts: LLaVa-1.6-7B [45, 43] and Open-LLaVa-Next-Llama3-8B [15]. We also include comparisons with frontier open-sourced models of similar size that are trained on larger datasets, namely LLaVa-OneVision-7B [36], Qwen2.5-VL-7B [5], and InternVL-38B [85].

LaViDa demonstrates competitive performance across a wide range of tasks spanning General, Reasoning, OCR, and Science categories. In general vision-language understanding, LaViDa-L achieves the highest score on MMMU [80] (43.3), outperforming all comparable models. LaViDa-D also ranks second on several benchmarks in this category. For reasoning tasks, both models surpass similarly scaled baselines on math-heavy and spatially grounded benchmarks. In Science, LaViDa achieves the best and second-best scores on ScienceQA [52] (81.4 and 80.2, respectively) while performing on par with Open-Llava-Next on AI2D [32], a complex diagram-based benchmark. Finally, in OCR LaViDa shows competitive performance but lags behind some of the latest AR models. This gap is primarily due to our use of average pooling for vision token compression, which leads to the loss of fine-grained spatial information. While this was a necessary trade-off given our limited compute budget, it poses challenges for tasks requiring precise text recognition and layout understanding. These results highlight the strength of LaViDa, demonstrating that diffusion-based approaches can scale competitively with AR models while achieving robust performance across a wide range of vision-language tasks.

Table 1: **LaViDa's Performance on Visual Understanding Tasks.** We report results on General, Reasoning, OCR, and Science Benchmarks. Dashes (–) denote results not reported. Open-Lnxt: Open-LLaVA-Next-Llama-3-8B; L-OV: LLaVA-OneVision-7B; Qwen2.5: Qwen2.5-VL-7B; Intern3: Intern-VL3-8B.

| | LaViDa-L | LaViDa-D | LLaVa-1.6 | Open-Lnxt | L-OV | Qwen2.5 | Intern3 |
|---|---|---|---|---|---|---|---|
| #Params | 8B | 7B | 7B | 8B | 7B | 7B | 8B |
| #Images (Pretrain) | 0.6M | 0.6M | 0.6M | 0.6M | 0.6M | >7M | - |
| #Images (SFT) | 1.0M | 1.0M | 0.7M | 1.0M | 7.2M | ~2M | 21.7M |
| *General* | | | | | | | |
| MME-P [23] | 1365.6 | 1463.5 | 1519.3 | **1610.9** | 1580.0 | - | - |
| VQAv2 [24] | 72.2 | 75.2 | **80.1** | 71.9 | - | - | - |
| MMBench [47] | 70.5 | 73.8 | 54.6 | **74.4** | 80.8 | 83.5 | 83.4 |
| MMMU [80] | **43.3** | 42.6 | 35.1 | 37.4 | 48.8 | 58.6 | 65.6 |
| *Reasoning* | | | | | | | |
| MME-C [23] | 341.1 | **378.9** | 322.5 | 336.8 | 418.0 | - | - |
| MathVista [51] | **44.8** | 42.1 | 34.4 | 41.8 | 63.2 | 68.2 | 75.2 |
| MathVerse [84] | **27.2** | 24.1 | 14.3 | 14.6 | - | - | - |
| MathVision [74] | **20.4** | 19.4 | 12.8 | 14.1 | - | - | - |
| *Science* | | | | | | | |
| ScienceQA [52] | 80.2 | **81.4** | 73.2 | 78.3 | 96.0 | - | - |
| AI2D [32] | 70.0 | 69.0 | 66.6 | **70.2** | 81.4 | 83.9 | 85.2 |
| *OCR* | | | | | | | |
| TextVQA [65] | 56.3 | 57.1 | **64.9** | 61.7 | - | 84.9 | 80.2 |
| DocVQA [55] | 59.0 | 56.1 | **74.4** | 69.9 | 87.5 | 95.7 | 92.7 |
| ChartQA [53] | 64.6 | 61.0 | 54.8 | **69.7** | 80.0 | 87.3 | 86.6 |
| InfoVQA [54] | 34.2 | 36.2 | **37.1** | 36.7 | 68.8 | 82.6 | 76.8 |

Table 2: **Performance of Specialized stage-3 Models.** We report (a) performance of LaViDa-Reason on math reasoning tasks and (b) performance of LaViDa-FIM on constrained poem completion task.

(a) Math benchmark results after long CoT distillation.

| | M.Vista↑ | M.Verse↑ | M.Vision↑ |
|---|---|---|---|
| LaViDa | 44.8 | 27.2 | 20.4 |
| LaViDa -Reason | **45.2** | **29.3** | **24.0** |
| Rel. Δ | +1% | +8% | +18% |

(b) Sentence and sample-level constraint satisfaction for poem completion tasks.

| | Sentence↑ | Sample↑ |
|---|---|---|
| LaViDa | **1.00** | **1.00** |
| LaViDa-FIM | **1.00** | **1.00** |
| LLaVa-1.6-7B | 0.41 | 0.37 |
| Qwen2.5-VL-7B | 0.45 | 0.16 |

## 4.3 Reasoning Distillation

Prior work has distilled LLMs [70] and VLMs [18, 73] using long chain-of-thought (CoT) data to elicit strong reasoning capabilities [25]. In the same spirit, we study the reasoning abilities of LaViDa by conducting additional stage-3 training using 19.2K CoT examples distilled from VL-Rethinker-7B, a strong reasoning model. We refer to the finetuned model as LaViDa-Reason. We evaluate it on MathVista [51], MathVerse [84], and MathVision [74] with CoT generation, comparing against the stage-2 results without CoT. We set the maximum generation length $L = 1024$ for these tasks. We report these results in Table 2a. We find that LaViDa-Reason outperforms LaViDa across all benchmarks, with the most significant gains observed on the most challenging MathVision reasoning dataset (+18% relative improvement). Further details are provided in Appendix B.3.

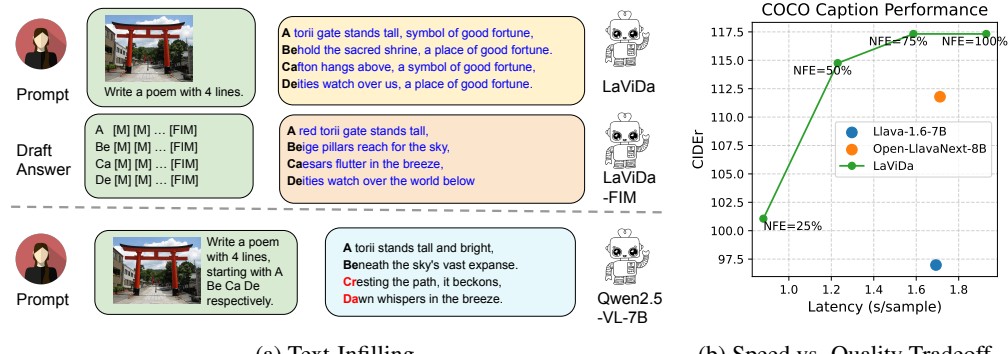

| (a) Text-Infilling | (b) Speed vs. Quality Tradeoff |

Figure 4: **We showcase the advantages of LaViDa over AR VLMS in terms of controllability and speed.** (a) Qualitative comparison on constrained poem generation between LaViDa /LaViDa-FIM and AR models. LaViDa variants successfully satisfy line-level constraints and adapt token length per line, unlike AR baselines. (b) Speed–quality tradeoff for image captioning on COCO 2017. By adjusting the number of discretization steps ($K$), LaViDa offers a tunable balance between latency and output quality (CIDEr score).

## 4.4 Text Infilling

LaViDa offers strong controllability for text generation, particularly in text infilling. Given a draft of $L$ tokens containing $L_M$ masks, we jump to timestep $t = \frac{L_M}{L}$ and run standard inference to reach $t = 0$. This directly replaces $L_M$ masks with $L_M$ tokens. However, in practice, the intended completion may require fewer tokens—e.g., "There is a [M][M][M][M] in the image" might become either "dog" or "traffic light". To allow variable-length completions, we conduct an additional stage-3 training using a 20% subset of stage-2 data and refer to this model as LaViDa-FIM. During training, we insert random-length [S]...[S][FIM] sequences mid-text. At inference, we append [FIM] to masked segments (e.g., [M][M][M][M][FIM]) to signal flexible termination. The model can then generate completions like [dog][S][S][S][FIM] or [traffic][light][S][S][FIM].

While FIM objectives are often discussed in the context of language tasks (e.g., code completion) [63, 7], they are equally relevant to multimodal applications. Figure 4a shows qualitative results on constrained poem generation, where the models generate a poem describing an image, with each line starting with specific syllables. Both LaViDa and LaViDa-FIM complete the task successfully, unlike AR models. Notably, LaViDa-FIM adapts token counts per line. Table 2b shows quantitative results over 100 samples: both LaViDa variants achieve 100% constraint satisfaction, while AR baselines remain below 50%. Additional results on other text-infilling use cases are provided in Appendix B.2.

## 4.5 Speed vs. Quality Trade Off

LaViDa offers a convenient way to achieve speed-quality tradeoffs by controlling the number of discretization steps $K$. We compare the performance on image captioning with 500 images from the COCO 2017 val dataset [42] with varying $K$. We set the maximum generation length to 32, and experimented with $K \in \{32, 24, 16, 8\}$, or equivalently, NFE$\in \{100\%, 75\%, 50\%, 25\%\}$. We report the average latency per image measured on a single A5000 GPU and the CIDEr score in Figure 4b. At NFE=$100\%$, LaViDa achieves a higher CIDEr score than AR baselines but is slightly slower. At NFE=$75\%$ and NFE=$50\%$, LaViDa is faster than the AR baselines and achieves better quality. At NFE=$25\%$, it is significantly faster but trails in performance. This indicates that LaViDa can flexibly adjust its inference speed based on application needs—allowing users to trade off generation latency and output quality depending on their specific requirements.

**Effect of KV Cache.** The speed of LaViDa relies on our proposed Prefix-DLM setup, which allows us to cache the keys and values of visual and prompt tokens [3]. In Table 3a, we compare the speed and sample quality between the proposed Prefix-DLM step and the uncached full-attention-mask step used in prior works like LLaDa. We find that Prefix-DLM significantly reduces latency and achieves a maximum speedup of $3.9\times$, with marginal performance cost. These experiments are performed using our stage-2 model, which is trained on a full-attention mask. We discuss training with customized

masks and customized kernels in Appendix B.4. In short, we found that these alternatives lead to considerable training overhead while offering little benefit.

Table 3: **Speed-Quality Tradeoff.** (a) We compare COCO image captioning performance with and without Prefix-DLM caching. Latency is measured in seconds/sample. (b) We compare the performance of different schedules at different NFEs.

<table>
<tr><td colspan="4" align="center">(a) Effect of KV Cache</td><td colspan="5" align="center">(b) Effect of timestep shifting</td></tr>
<tr><td rowspan="2">Method</td><td rowspan="2">NFE</td><td rowspan="2">CIDEr ↑</td><td rowspan="2">Latency↓</td><td colspan="5" align="center">COCO Caption (CIDEr)↑</td></tr>
<tr><td>NFE</td><td>25%</td><td>50%</td><td>75%</td><td>100%</td></tr>
<tr><td>Full-DLM</td><td>100%</td><td>**121.0**</td><td>7.65</td><td>cosine</td><td>87.7</td><td>102.2</td><td>110.8</td><td>117.3</td></tr>
<tr><td>Prefix-DLM</td><td>100%</td><td>117.3</td><td>1.93</td><td>linear</td><td>84.9</td><td>105.2</td><td>108.6</td><td>117.3</td></tr>
<tr><td>Full-DLM</td><td>50%</td><td>118.6</td><td>4.13</td><td>$\alpha$=3</td><td>48.7</td><td>74.7</td><td>92.4</td><td>117.3</td></tr>
<tr><td>Prefix-DLM</td><td>50%</td><td>114.8</td><td>**1.23**</td><td>$\alpha$=$3^{-1}$</td><td>**101.1**</td><td>**114.8**</td><td>**117.3**</td><td>**117.3**</td></tr>
<tr><td>Open-Lnxt-8B</td><td>–</td><td>111.8</td><td>1.71</td><td></td><td></td><td></td><td></td><td></td></tr>
</table>

**Noise Schedule.** To study the effect of the proposed time-step shifting schedule, we compare the performance of different schedules with NFE$\in \{100\%, 75\%, 50\%, 25\%\}$. We compare the proposed time-step shifting with $\alpha = 3$, and $\alpha = 3^{-1}$, as well as linear and cosine schedule baselines. We report results on COCO image captioning [42] in Table 3b. The convex schedule with $\alpha = 3^{-1}$ works the best. We also observe similar behaviors when conducting CoT inference using LaViDa-Reason on MathVision [74] dataset. At NFE=50%, $\alpha = 3^{-1}$ achieves an accuracy of 21.05, which is 30% higher than 16.12 achieved by a linear schedule. We provide results on MathVision in Appendix B.3.

## 4.6 Ablation Studies

We conducted a thorough ablation of various design choices. In the main paper, we discuss the effect of complementary masking and input image resolution. We provide further discussion of other design choices in the Appendix B.5. We conducted these experiments using a 200k subset of our (stage-2) training data. We report results in Tables 4a and 4b respectively. Table 4a shows that our proposed **complementary masking** scheme leads to considerable improvements across all benchmarks, most notably, complementary masking leads to a relative improvement of 67% on ScienceQA [52] with affordable compute overhead during the training (8% slowdown). Table 4b shows that **high-resolution input** improves overall performance, with the gain on OCR tasks being more pronounced than generic vision tasks (e.g., VQAv2). We did not use average pooling for the low-resolution setup. We provide additional details in the Appendix B.

Table 4: **Ablation Studies.** We study the effect of (a) complementary masking and (b) image resolution. For (a), we also report the wall clock time of 1000 training steps with a batch size of 128.

<table>
<tr><td colspan="3" align="center">(a) Effect of complementary masking.</td><td colspan="3" align="center">(b) Effect of image resolution.</td></tr>
<tr><td></td><td>w/o Comp.M.</td><td>w/ Comp.M.</td><td></td><td>$384^2$</td><td>$768^2$</td></tr>
<tr><td>MME↑</td><td>260.00</td><td>**297.00**</td><td>TextVQA↑</td><td>48.40</td><td>**55.65**</td></tr>
<tr><td>MathVista↑</td><td>28.40</td><td>**33.40**</td><td>DocVQA↑</td><td>43.22</td><td>**58.72**</td></tr>
<tr><td>ScienceQA↑</td><td>48.74</td><td>**81.49**</td><td>ChartQA↑</td><td>42.20</td><td>**57.70**</td></tr>
<tr><td>MMMU↑</td><td>38.56</td><td>**41.78**</td><td>InfoVQA↑</td><td>26.48</td><td>**36.23**</td></tr>
<tr><td>Runtime↓</td><td>**8.2 hr**</td><td>8.9 hr</td><td>VQAv2↑</td><td>65.92</td><td>**66.78**</td></tr>
</table>

## 5 Conclusion

In conclusion, we propose LaViDa, the first family of vision-language models based on DMs. To address various challenges, we introduce several novel training and inference techniques, including complementary masking, Prefix-DLM cache, and timestep shifting. Through extensive experiments, we show that these techniques significantly outperform a naive adaptation of DMs for visual tasks.

Using a comprehensive evaluation suite, we demonstrate that LaViDa achieves competitive performance compared to AR models trained under similar settings, while offering unique advantages such as speed–quality tradeoffs and controllability via text infilling. Our work proves that LaViDa can be a powerful alternative to exitsing AR VLMs, extending the prior success of DMs in the language domain to the vision space.

## 6  Acknowledgement

AG would like to acknowledge support from NSF CAREER Grant #2341040, Schmidt Sciences Early Career Fellowship, and Amazon Research Award.

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

# A Additional Technical Details

## A.1 Formulation of Discrete Diffusion Models (DMs)

In this section, we provide a detailed review of the formulation of Discrete Diffusion Models (DMs) briefly described in 2.2. Given a text sequence of $L$ tokens $X_0 = [X_0^1, X_0^2, ...X_0^L]$, the forward process $q(X_t|X_s)$ gradually coverts it to a sequence of mask tokens $[M]$, denoted by $X_1 = [X_1^1, X_1^2, ...X_1^L]$ over the continuous time-interval $[0, 1]$, with $1 \geq t \geq s \geq 0$. Formally, this process is defined as

$$q(X_t^i|X_s^i) = \begin{cases} \text{Cat}(X_t^i; \mathbf{M}), & \text{if } X_s^i = M \\ \text{Cat}(X_t^i; \frac{1-t}{1-s}\mathbf{X_s^i} + \frac{t-s}{1-s}\mathbf{M}), & \text{if } X_s^i \neq M \end{cases} \tag{4}$$

which has the marginal

$$q(X_t^i|X_0^i) = \text{Cat}(X_t^i; (1-t)\mathbf{X_0^i} + t\mathbf{M}) \tag{5}$$

where Cat(.) denotes a categorical distribution and $\mathbf{M}, \mathbf{X_0^i}, \mathbf{X_s^i}$ are probability vectors. MDLM [64] showed that the posterior of the reversal process $p(X_s|X_t, X_0)$ can be simplified into the following

$$p(X_s^i|X_t^i, X_0^i) = \begin{cases} \text{Cat}(X_s^i; \mathbf{X_t^i}), & \text{if } X_s^i \neq M \\ \text{Cat}(X_t^i; \frac{t-s}{t}\mathbf{X_0^i} + \frac{s}{t}\mathbf{M}), & \text{if } X_s^i = M \end{cases} \tag{6}$$

In practice, we use the categorical distribution induced by the neural network's prediction $p_\theta(X_0^i|X_t)$ in place of $\mathbf{X_0^i}$ the sample from the reverse process, which gives the following parametrization

$$p_\theta(X_s^i|X_t) = \begin{cases} \text{Cat}(X_s^i; \mathbf{X_t^i}), & \text{if } X_s^i \neq M \\ \text{Cat}(X_t^i; \frac{t-s}{t}p_\theta(X_0^i|X_t) + \frac{s}{t}\mathbf{M}), & \text{if } X_s^i = M \end{cases} \tag{7}$$

**Inference Algorithm.** Given a target length $L$ and discretization steps $t_0, t_1...t_K$ where $t_0 = 0$ and $t_K = 1$, we first initialize $X_{t_K}^{1:L} = X_1^{1:L} = M$, then use Equation 7 to repetitively sample $p_\theta(X_{t_{k-1}}|X_{t_k})$ until we reach $t_0 = 0$. In this process, we also assume $p_\theta(X_{t_{k-1}}|X_{t_k})$ factorize into $\prod_{i=1}^L p_\theta(X_{t_{k-1}}^i|X_{t_k})$ following previous works [57, 64, 49].

**Training Objective.** Recall that the training objective of DMs introduced in Section 2.2 is formulated as

$$\mathcal{L}_{\text{DLM}} = -\mathbb{E}_{t,X_0,X_t} \left[ \frac{1}{t} \log p_\theta(X_0|X_t) \right] \tag{8}$$

where $p_\theta(X_0|X_t)$ factorizes into $\prod_{i=1}^L p_\theta(X_0^i|X_t)$. However, Equation 6 shows that $p_\theta(X_0^i|X_t)$ has a closed form solution that depends only on $X_t$ when $X_t^i \neq M$. Intuitively, this comes from the fact that in the reverse process, once a token $X_i$ changes from a mask token to a clean text token, it stays the same thereafter. Based on this observation, we can remove the terms that does not depend on the neural network $\theta$ from the learning objective, giving us the following loss

$$\mathcal{L}_{\text{DLM}} = -\mathbb{E}_{t,X_0,X_t} \left[ \frac{1}{t} \sum_{X_t^i \neq M} \log p_\theta(X_0^i|X_t) \right] \tag{9}$$

Hence, the loss is only computed over the masked indices in $X_t$.

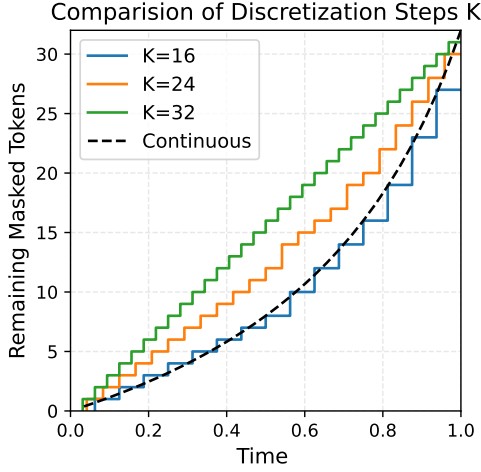 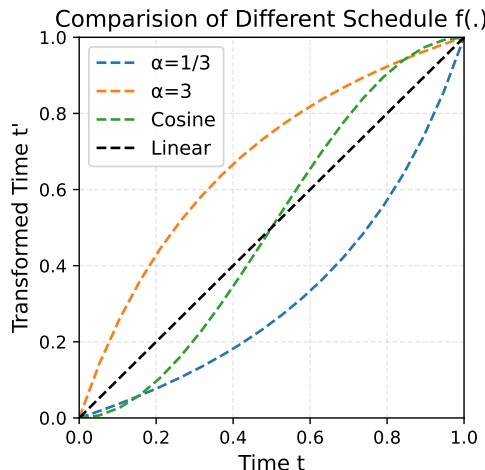

| (a) Discretized Schedules at Different $K$. | (b) Different Choice of Continuous Schedules $f$. |

Figure 5: **Visualization of Schedules** (a) We visualize discretization results of the same continuous schedule (dashed line) under different numbers of sampling steps $K \in \{16, 24, 32\}$ at $L = 32$. (b) We visualize various choices of continuous schedules: Shift($\alpha = 3^{-1}$), Shift($\alpha = 3$), Cosine, and Linear.

## A.2 Timestep shifting Schedule

Given a sequence of $L$ tokens, and a timestep $t \in [0, 1]$, in expectation $X_t$ contains $tL$ masked tokens. In practice, we implement Equation 7 such that there are exactly $\lfloor tL \rfloor$ masked tokens and $L - \lfloor tL \rfloor$ clean text tokens at timestep $t$ during the sampling process.

Given a fixed number of sampling step $K$, we define the canonical discretization as $t_i = \frac{i}{K}$ for $i = 0, 1..K$, with $t_0 = 0$ and $t_1 = 1$. This forms a uniformed sampling schedule, where roughly a fixed mount of $\frac{L}{K}$ tokens is unmakes at each sampling step. Any other schedule $t'_i$ can be defined as $t'_i = f(t_i)$ where $f(.)$ is a monotonic function such that $f(0) = 0$ and $f(1) = 1$.

When $f(.)$ is convex, the slope will be steeper when $t$ get closer to 1 , indicating that more tokens are decoded in earlier sampling steps. By contrast, when $f(.)$ is concave, the slope will be steeper when $t$ get closer to 0, indicating that more tokens are decoded in later sampling steps.

**Choice of Schedule.** We explored a wide range of choices for the continuous schedule $f(.)$. The *timestep shifting schedule* is a family of schedule defined as

$$f(t) = s_\alpha(t) = \frac{\alpha t}{1 + (\alpha - 1)t} \tag{10}$$

where $\alpha$ is a hyperparameter. When $\alpha < 1$, the schedule is convex. When $\alpha > 1$, the schedule is concave. The *cosine schedule* is defined as

$$f(t) = 1 - \frac{1}{2}(1 + \cos(\pi t)) \tag{11}$$

The *linear schedule* is just the identity function $f(t) = t$. We visualize these choices in Figure 5b.

**Rounding in Discretization.** In principle, we can pick any $f(.)$. However, given a particular choice of $L$ and $K$, if $\lfloor f(t_k)L \rfloor$ and $\lfloor f(t_{k-1})L \rfloor$ yields the same integer, then no tokens are unmasked when we compute $p_\theta(X_{t_{k-1}}|X_{t_k})$. Hence, the actual number of function calls to the model $\theta$ may be less than, $K$ depending on the choice of $f(.)$. To make the sampling compute cost more predictable and allow for a fair comparison across different schedulers, we augment all choices of $f(.)$ to $f^K(.)$ such that $\lfloor f^K(t_{k-1})L \rfloor < \lfloor f^K(t_k)L \rfloor$ (i.e. at least one token is decoded at each step). Note that in the sampling process, the exact real value of $f^K(t_k)$ does not matter as long as it does not change

$\lfloor f^K(t_k)L \rfloor$. Hence, we can parameterize the sampling process in an alternative manner using a sequence of integers $F_k^K = \lfloor f^K(t_k)L \rfloor$, with $F_0^K = 0$ and $F_1^K = L$. Formally, we set $F^K$ by solving the following optimization objective

$$\min_{\{F_{0:K}^K\} \subset \{0,1..L\}} \sum_{k=0}^{K} \left\| F_k^K - f(t_k)L \right\|^2$$
$$\text{subject to} \quad F_{k-1} < F_k, \quad \text{for } k = 1, 2, \ldots, K$$
$$F_0^K = 0$$
$$F_1^K = L$$

We visualize such examples in Figure 5a. We set $L = 32$ and $K \in \{16, 24, 32\}$. When $K = 32$, $F^K$ is effectively a linear schedule, since the only schedule with 32 steps that satisfy the constraint $F_{k-1} < F_k$ is a uniform schedule where we unmask exactly one token per step. As $K$ reduces to 24 and 16, we see the discretized schedule becomes closer to the continuous scheduler (visualized in dashed line).

### A.3  Padding

We follow the design of LLaDa [57] and apply the loss function to both standard text tokens and padding tokens. AR models typically do not compute the loss over padding tokens. However, when sampling from DMs, we have a specified generation length $L$. In the generation process, we unmask $L$ mask tokens to $L$ non-mask tokens. Since the length of the desired answer may not be exactly $L$ tokens, the model will generate padding tokens. To achieve this capability, we pad the sequences during the training, and apply the loss on the padding tokens following LLaDa.

## B  Additional Experiment Details and Results

### B.1  Setup

In this section, we document the detailed training setup, including data, hyperparameters and compute used for the main experiments. Additional details about Prefix-DLM Cache can be found in B.4. Additional details about stage-3 math reasoning experiments can be found in B.3.

**Training Data Composition.** For the pretraining phase, we use LCS-558K [46] consists of 558K image-text pairs. For the finetuning phase, we mostly use the dataset released by Open-LLaVa-Next [15]. We made some small adjustments to the weight of each data source and increased the weight of some QA dataset. This is used to compensate the fact that our model only learns from a randomly chosen round at each training step for multi-round QA data. We document the precise dataset composition of our stage-2 training in Table 5.

Table 5: **Compositio of Stage-2 Training Data**. We report the data sources and sample sizes used to compose the Stage-2 finetuning data.

| Data Source | Size | Data Source | Size | Data Source | Size |
|---|---|---|---|---|---|
| COCO[42] | 349,860 | GQA[30] | 72,140 | DocVQA[55] | 10,211 |
| ALLaVA-VFLAN[12] | 202,966 | Synthdog-En[33] | 29,765 | DVQA[31] | 10,000 |
| Visual Genome[35] | 86,417 | TextVQA[56] | 21,953 | SA-1B[34] | 8,997 |
| OCR VQA[56] | 80,000 | ChartQA [53] | 18,317 | LLaVA-150K [46] | 2,988 |
| GeoQA+ [13] | 72,318 | AI2D[32] | 12,413 | WikiArt[67] | 500 |
| Share-TextVQA [14] | 500 | Web-Celebrity[15] | 500 | Web-Landmark[15] | 500 |

**Training Hyperparameters.** We use AdamW optimizer with a learning rate of 5e-3 with a cosine decay schedule for all experiments. For pretraining (Stage 1), we adopted a global batch size of 256 and trained for 1 epoch. For finetuning (Stage 2), we adopted a global batch size of 512 and trained for two epochs.

**Compute Used.** We used a mixture of A100s and A6000s for training experiments and A5000 for evaluations and inference speed benchmarks. Because of the memory constraint, we set the per GPU batch size to 8 on A100s and 4 on A6000s. We adjust the gradient accumulation steps accordingly so that the global batch size is always 256 for the pretraining and 128 for the finetuning stage.

**Evaluation Setup.** We implement our evlaution using `LMMS-Eval` [83] library. We use the default prompt provided by the library for all benchmarks. We report the split used and the generation length $L$ in Table 6.

Table 6: **Evaluation Setup.** We report evaluation split and generation length $L$ used to produce results of Table 1 in the main paper. *We use a generation length of 100 for LaViDa and a generation length of 1024 for LaViDa-Reason.

| Dataset | Split | $L$ | Dataset | Split | $L$ |
|---------|-------|-----|---------|-------|-----|
| MME-P | test | 100 | MathVista* | testmini_format | 100 |
| VQAv2 | val | 16 | MathVerse* | testmini_vision_dominant | 100 |
| MMBench | dev | 100 | MathVision* | mathvision_testmini | 100 |
| MMMU | val | 16 | ScienceQA | scienceqa-full | 16 |
| MME-C | test | 16 | AI2D | test | 16 |
| TextVQA | val | 16 | ChartQA | test | 16 |
| DocVQA | test | 32 | InfoVQA | test | 32 |

## B.2 Text-Infilling

In this section, we provide additional qualitative results of the text-infilling capability of LaViDa. We visualize the results in Figure 6. In the first example, we ask the model to extract multiple attributes from the image in JSON format. In the second example, we ask the model to edit a sentence based on the image. This is achieved by deleting the original sentence and inserting mask tokens. In the final example, we ask the model to complete a movie script based on the image prompt. LaViDa was able to successfully complete these tasks.

While it may be possible to achieve similar results using an AR model, they require careful prompting. By contrast, as shown in Figure 6, using a diffusion model for text-infilling is more straightfoward.

## B.3 Math Reasoning

In this section, we provide additional training setup for LaViDa-Reason and provide additional experiment results on the speed-quality tradeoff on math reasoning.

**Data and Training Setup.** In §4.3, we train LaViDaon long chain-of-thought (CoT) data to get LaViDa-Reason. Specifically, we choose a strong 7B reasoning model, VLRethinker-7B [73] as a teacher model to generate the long reasoning traces. Further, we choose their own ViRL-39K data that contains (image, question, final answer). Subsequently, we generate the CoT and predicted final answer from the teacher model and filter the ones that lead to the correct final answer [81, 6]. This led to the creation of the final dataset of size $19.2K$.[1] In particular, we finetune LaViDaon this data for 5 epochs using the identical training setup as stage-2 (e.g., batch size, learning rates) and chose the checkpoint that achieves the best performance on MathVision (testmini). We observe that the same checkpoint achieved a good performance on the MathVerse and MathVista dataset too. During inference, we set the generation length to 1024 since LaViDa-Reason synthesizes long chain-of-thoughts for problem-solving.

**Speed-Quality Tradeoff.** In the main paper, we reported the speed-quality tradeoff results on COCO image captioning and discovered that the convex schedule works the best. We conducted similar study on LaViDa-Reason for CoT inference on MathVision dataset. We report these results in Table 7. Overall, the conclusion on CoT math reasoning task is similar to that on image captioning task, with the convex schedule performing the best across different choices of sampling steps. To

---

[1]We will release this data in the camera-ready version.

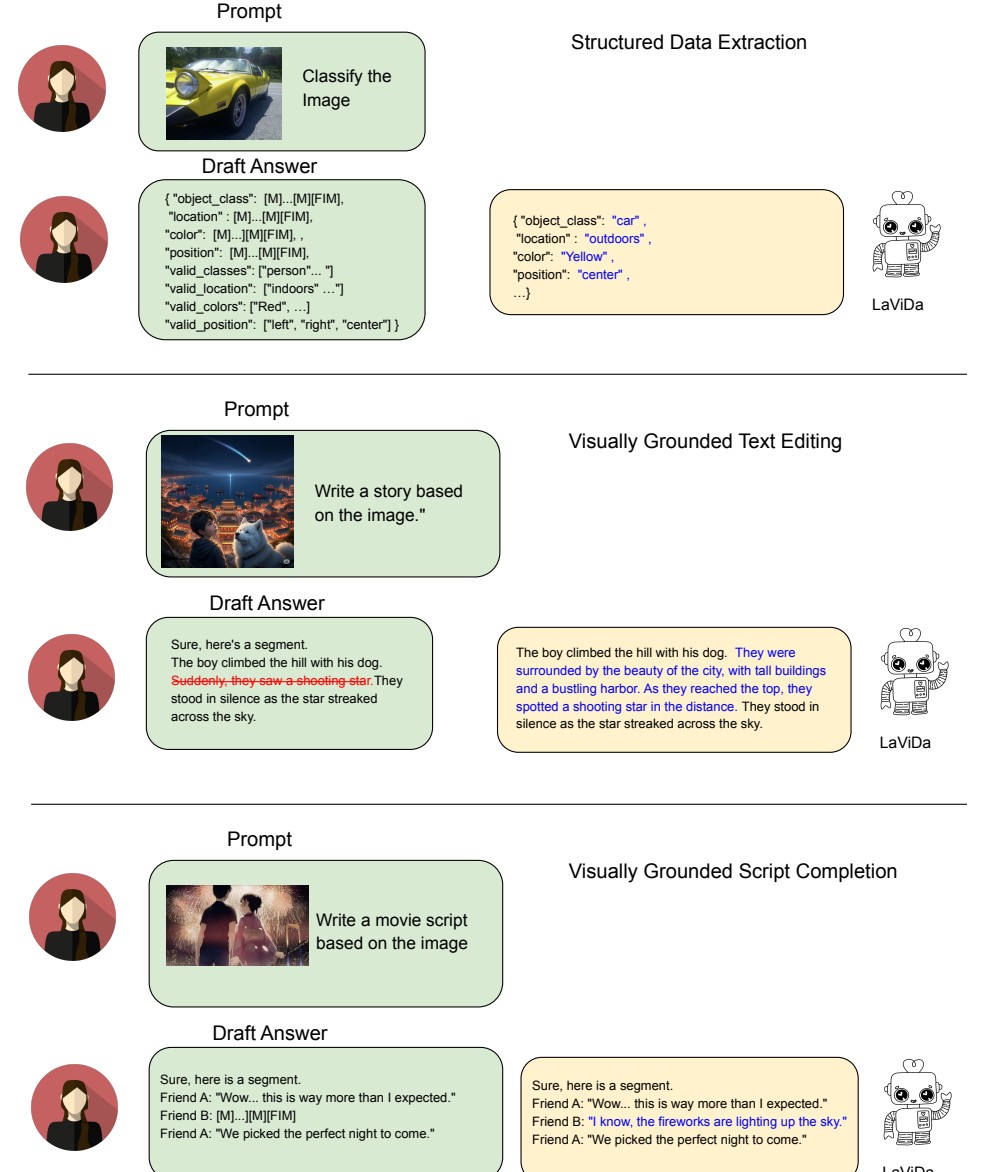

Figure 6: **Additional Qualitative Results for Text Infilling.** We showcase several useful applications of text-infilling capabilities.

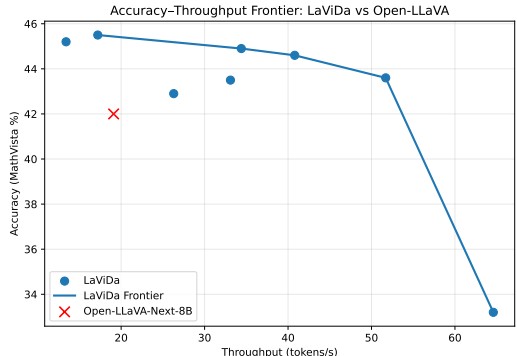

Figure 7: **Speed-Quality Tradeoff on MathVista Dataset.** We report the throughput and accuracy on MathVista dataset of LaViDa under a variety of inference setup.

further examine the speed-quality tradeoff, we visualize the inference throughput (tokens/s) and accuracy on MathVista dataset under different inference setup in Figure 7 and compare with Open-LLaVA-Next-8B baseline. LaViDa's frontier strictly dominates the performance of autoregressive Open-LLaVA-Next-8B both in terms of speed and quality.

Table 7: **Effect of different schedule on MathVision dataset.** We study the effect of different schedules on MathVision dataset using Chain-of-Though inference with LaViDa-Reason.

| NFE | MathVision Acc↑ | | |
|---|---|---|---|
| | 25% (256 steps) | 50% (512 steps) | 100% (1024 steps) |
| cosine | 8.55 | 13.49 | 24.02 |
| linear | 10.86 | 16.12 | 24.02 |
| $\alpha$=3 | 5.59 | 8.88 | 24.02 |
| $\alpha$=$3^{-1}$ | **12.5** | **21.05** | **24.02** |

## B.4    Prefix-DLM

In this section, we discuss several alternatives to our prefix-DLM setup that we explored and document the experiment results.

**Inference Algorithm.** Autoregressive models employ a causal attention mask. Because of this, they can leverage KV cache for effective inference. By contrast, discrete diffusion models (DMs) used a full attention mask. While DMs can decode multiple tokens in parallel, it cannot leverage attention mask for fast inference. Prefix-DLM combine the best of both worlds by introducing a prefix attention mask such that the queries of image embeddings and text prompts can only interact with keys and values of image embeddings and text prompts, but not the keys and values of answer tokens. Through this mechanism, we can leverage the KV cache for the image embeddings and text prompts. In vision-language applications with a long context (900+ vision tokens per image), this saves a lot of compute at inference time, while preserving the full bidirectional context.

A alternative to our prefix-DLM was the recently proposed semi-autoregressive Block-Diffusion [3], which uses a block-wise causal attention mask. In this setup, the input sequece are chunked into a sequence of fixed length blocks, each containing $L_B$ tokens. A token in Block $i$ can see all tokens in the past and current Block $j$ with $j \le i$, but cannot see all future blocks Block $j$ with $j > i$. While this design allows it to leverage block-wise KV cache, it limits the bi-directional context to at most $L_B$ tokens in the future, which is undesirable for tasks like text-infilling. Additionally, because of its semi-autoregressive nature, when we are generating Block $i$, we must see all mask-free past Blocks $j$ with $j < i$. Hence, the naive training algorithm can mask at most $L_B$ tokens in each training sample (i.e. the last block), which is inefficient. To address this issue, a customized attention kernel was developed to allow for parallel training. However, this leads to considerable training overhead. By contrast, Prefix-DLM can leverage the KV cache while having the full bidirectional context. It also

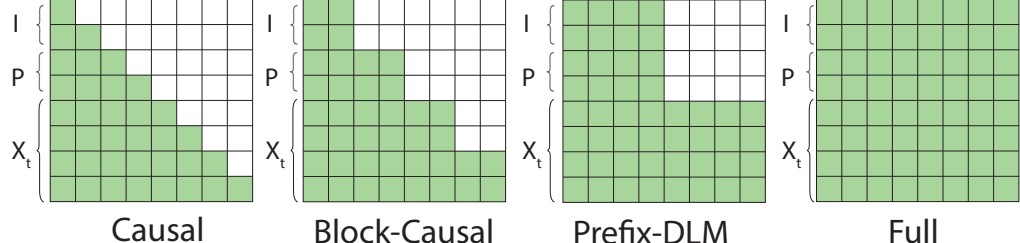

Figure 8: **Visualization of Different Choices of Attention Mask at Inference Time.** $I$ represents the image embeddings, $P$ represents the prompt tokens, and $X_t$ represents the partially masked answer tokens. Each row represents a query and each column represents a key. Colored region indicts tokens queries and keys can interact with each other.

does not require any specialized training algorithms, since we adopt it as a pure inference technique of DMs.

We visualize the different choices of attention masks in Figure 8. We also compare the properties of different choices in 8.

Table 8: **Comparison of Different Choices of Attention Mask**. We compare properties of different choices of attention masks. The desired properties are highlighted.

| Method | Attn. Mask | Bi-Direction Context | KV Cache | Training Overhead |
|---|---|---|---|---|
| AR | Causal | None | Yes | No |
| DM | Full | Full Seq. | No | No |
| Block-Diffusion | Block-Causal | Within Block | Yes | Yes |
| LaViDa | Prefix-DLM | Full Seq. | Yes | No |

**Training Algorithm.** We adopt Prefix-DLM as a pure inference algorithm. Our training process is identical to that of a standard DM with full attention mask. We made this choice mostly because of efficiency reasons. We also explored adopting the prefix-DLM attention mask during the training (Prefix-DLM-FT) with four different but equivalent implementations. Generally, these four implementations are categorized into two classes: Prefix-DLM-FT1 and Prefix-DLM-FT2. We visualize these setups in Figure 9.

Recall from Section 3.2 that we adopted a complementary masking scheme, wherein given each triplet of image $I$, prompt $P$ and answer $X$, we create two versions of partially masked answer $X_t$ and $X_t^C$ with complementary masking in order to utilize all tokens in the clean answer $X$. Empirically, this is achieved by copying the prompt embedding and concatenate $X_t$ and $X_t^C$ to different copies respectively (Left Column of Figure 9). By default, we use the full attention for both copies. We can adopt the Prefix-DLM attention mask during the training for each copy individually. We call this setup Prefix-DLM-FT1 (Middle Column of Figure 9). Alternatively, we can concatenate $I, P, X_t, X_t^C$ into a single long sequence and manipulate the attention mask such that queries of $I, P$ can see keys of $I, P$, queries of $X_t$ can see keys of $I, P, X_t$ and queries of $X_t^C$ can see keys of $I, P, X_t^C$. We call this setup Prefix-DLM-FT2 (Right Column of Figure 9).

For each of the two variant Prefix-DLM-FT1 and Prefix-DLM-FT2, we designed two concrete implementations. The first set of implementations (Prefix-DLM-FT1-Mask,Prefix-DLM-FT2-Mask) simply construct a 2D attention mask of size $L \times L$ for each sample, where $L$ is the sequence length, and pass it to the torch SDPA kernel. The second variant (Prefix-DLM-FT1-Flex,Prefix-DLM-FT2-Flex) merely constructs an integer tensor of size 3 per sample, containing the length of $I, P, X$. We then use a customized `flex_attention` module [20] to implement the attention mask implicitly based on the length of $I, P, X$.

Notably, these four variants of Prefix-DLM-FT generates identical loss value and will produce exactly the same training dynamics (disregarding small numerical differences between implementations). The only difference is the efficiency. Hence, we only benchmarked the performance and performed the full training using the most efficient version.

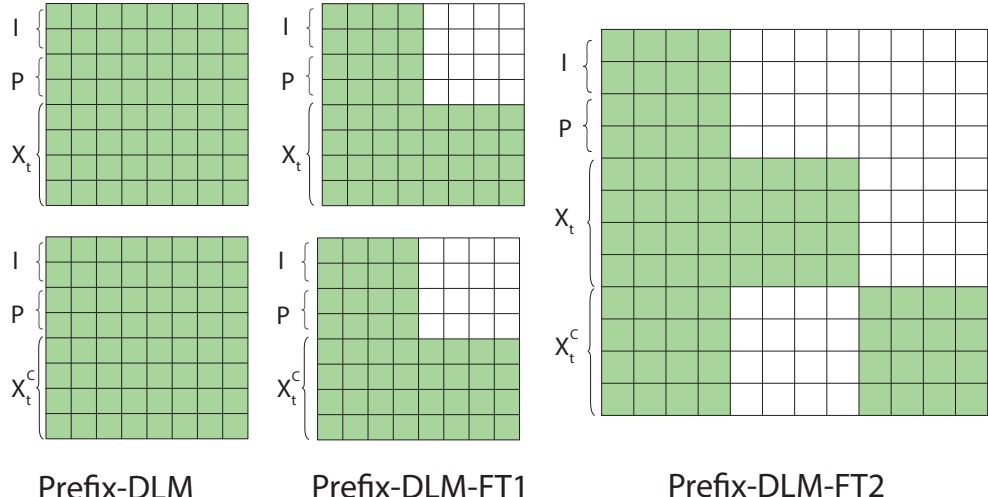

Figure 9: **Visualization of Training Strategies for Prefix-DLM.** Given each triplet of image $I$, prompt $P$ and answer $X$, we create two versions of partially masked answer $X_t$ and $X_t^C$ with complementary masking.(Left) By default, we construct two sequence and apply full attention mask. (Middle) Prefix-DLM-FT1 applies prefix attention mask to each copy independently. (Right) Prefix-DLM-FT2 combines $I, P, X_t, X_t^C$ into a single sequence and manipulate the attention mask to achieve an equivalent effect to Prefix-DLM-FT1.

We report the speed benchmark in Table 7. Overall, even the fastest finetuning version is 62% slower than the full attention mask baseline, suggesting a high overhead caused by batch-dependent masking strategies during training. We also report the model performance after 1,500 training steps (roughly 200k samples from the training data) in Table 10.

Overall, Prefix-DLM and Prefix-DLM-FT has mostly identical performance, with Prefix-DLM having a small lead over many tasks. Because of this, we consider the 62% overhead as unacceptable and opt for a training procedure using the full attention mask. This also gives user an additional dimension for speed-quality tradeoff: they can disable Prefix-DLM cache and use the full attention mask at inference to achieve a slightly better performance (Results shown in Table 3a in main paper).

Table 9: **Training Speed of Different Variants of Prefix-DLM.** We report the average training speed of different setup with a batch size of 128 on 8 A6000 GPUs.

| Method | Speed(s/training step) |
|---|---|
| Prefix-DLM | **37.2** |
| Prefix-DLM-FT1-Mask | 67.3 |
| Prefix-DLM-FT2-Mask | 82.6 |
| Prefix-DLM-FT1-Kernel | 60.2 |
| Prefix-DLM-FT2-Kernel | 74.4 |

Table 10: **Performance of Prefix-DLM and Prefix-DLM-FT across benchmarks.** We compare the performance of two variants after 1,500 steps of training.

| | MMMU | VQAv2 | MME | M.Vista | M.Verse | ScienceQA | AI2D |
|---|---|---|---|---|---|---|---|
| Prefix-DLM | **40.56** | **63.26** | 286.79 | 33.60 | **19.67** | **80.36** | **64.31** |
| Prefix-DLM-FT | 40.10 | 61.02 | **290.71** | **35.00** | 18.15 | 80.15 | 64.15 |

Table 11: **Ablation Studies on the Choice of Vision Encoders.** We compare the performance of models with different vision encoders after 1,500 steps of training. On Average, SigLip improves over CLIP by **+1.86** and improves over MetaCLIP by **+2.77**.

|          | MMMU      | VQAv2     | MME        | M.Vista   | M.Verse   | ScienceQA | AI2D      |
|----------|-----------|-----------|------------|-----------|-----------|-----------|-----------|
| SigLip   | **40.56** | **63.26** | 286.79     | 33.60     | 19.67     | **80.36** | **64.31** |
| CLIP     | 40.44     | 59.58     | **288.21** | 30.90     | **20.05** | 76.56     | 59.78     |
| MetaCLIP | 40.33     | 55.96     | 280.36     | **33.80** | 17.26     | 78.66     | 62.79     |

## B.5  Ablation Studies

We conduct additional ablation studies over the choice of vision encoders. We experimented with SigLip[82], CLIP [60], and MetaCLIP[76]. We report the model performance after 1,500 training steps (roughly 200k samples from the training data) in Table 11. Overall, SigLip achieves the strongest overall performance, with notably gains in VQAv2, ScienceQA, and AI2D.

## B.6  Additional Scaling

While LaViDa outperforms AR VLMs under comparable data scale, there is still a considerable gap between LaViDa and state-of-the-art AR VLMs. To validate the scalability of LaViDa, we scale the training schedule by 2x, resulting in considerable performance on general understanding and OCR tasks. We report these results in Table 12. Due to compute constraints, we left further scaling to future works.

Table 12: **Additional Scaling Experiments on OCR and General Understanding Tasks.**

| Model               | MME       | MMBench  | ChartQA  | DocVQA   | InfoVQA  |
|---------------------|-----------|----------|----------|----------|----------|
| LaViDa              | 341.1     | 70.5     | 64.6     | 59.0     | 34.2     |
| +Additional Training| **444.3** | **75.6** | **74.9** | **68.6** | **43.4** |
| Open-LLaVA-Next-8B  | 336.8     | 74.4     | 69.7     | 69.9     | 36.7     |

# C   Additional Backgrounds

In this section, we provide additional discussions with relevant works not covered in the main paper.

**Masked Generative Models.** Masked generative modeling has a long history before the recent advancements of DMs. Earliest works such as BERT[19] and MAE[27] adopt the masked generative modeling objective as a pretraining objective to learn rich text and vision features. They mainly concern the utility of learnt features to downstream perception and understanding tasks, instead of the generation capability of the model. A series of subsequent works use mask modeling to build generative models for images [11, 10], texts [40] and audio [86]. Compared with these early works relying on ad-hoc sampler designs, recent works on DMs [49, 64] provided a principled way for training and sampling from masked generative models.

**Multi-Modal Diffusion Models.** Several works have explored to build a multi-modal diffsion models with vision-language capabilities. CoDi [68] and OmniFlow [38] build continuous diffusion model over a latent text embedding space and use autoregressive decoder to decode the generated latent mebeddings to actual texts. UniD3 [29] and UniDisc [66] build discrete diffusion models for simultaneous text and image generation. Overall, these models have limited language generation capabilities. Their experiments on text generations are limited in scale and mostly focusing on captioning. They cannot perform more complex instruction-following and understanding tasks (e.g. reasoning) like many modern autoregressive VLMs.

# D   Limitations

In this section, we discuss the limitations of LaViDa. There are mainly two limitations.

**Scale.** While LaViDaachieves competitive results when compared against similar-sized AR VLMs trained on a similar scale of data, there remain a considerable gap between LaViDa's performance and that of state-of-the-art open sourced VLMs such as LLaVa-OneVision and Qwen2.5-VL. These models either comes with more training data or larger model sizes. Future work should study if DMs scale well with a larger model and more training data.

**OCR.** LaViDa's performance on OCR tasks are slightly worse than the baselines, this can be mainly attributed to the average pooling operation which we introduced to reduce the sequence length by compressing the visual information. Concretely, LLaVa-1.6-7B and Open-LLaVa-Next-Llama3-8B baseline do not adopt average pooling, and uses 2880 tokens ($24 \times 24 \times 5$ Views) to represent each image. In our setup, removing the average pooling would lead to a total of 3645 tokens ($24 \times 24 \times 5$ Views) per image. Average pooling is necessary because the base model LLaDa and Dream has a context length of 4096 and 2048 respectively. Without average pooling, LLaDa will not have enough context length to fit longer training samples, while Dream will not have enough context length to even fit one image.

We also tried to extend the context length of these models via techniques such as rope rescaling, but achieved limited success. We experimented with extending Dream to 4096 context length and evaluate the model using the needle-in-a-haystack task [28] at 4096 context length. We found that Dream-7B (72.4 Acc) performs worse than LLaDa-8B (91.6 Acc) and Llama-3-8B (95.4 Acc).

We hope future works on DMs with longer context will provide stronger base models to finetune on vision-language applications.

# E  Boarder Impacts and Safeguards

Our model may inherit the biases embedded in the base model LLaDa-8B and Dream-7B, as well as biases incorporated in the training data. Our model may also suffer from the same hallucination problem as the base model. Just as any LLMs and VLMs, there is also the risk that our model is used to generate harmful or offensive content. Overall, our model is intended to be used by researchers to build a strong diffusion model for vision-language applications. We do not recommend it be used for any other purposes.

# F  License Information for Data and Models

We report the licenses of the used artifacts in Table 13. We followed the intended use of all respective artifacts.

Table 13: **Licenses and sources for datasets and models used.** Datasets marked with * have no published license; the accompanying paper's arXiv license is used as fallback. Training entries marked with † or ‡ reflect compound or inferred licensing conditions.

| Category | Name | License | Platform |
|---|---|---|---|
| Vision Encoder | SigLIP | Apache 2.0 | Hugging Face |
| Language Model | LLaDA-8B | MIT | Hugging Face |
| Language Model | Dream-7B | Apache 2.0 | Hugging Face |
| VLM | LLaVA-NeXT (1.6) | Apache 2.0 | GitHub |
| VLM | LLaVA-NeXT (1.6) | LLaMA 2 License | Hugging Face |
| VLM | Open-LLaVA-NeXT | Apache 2.0 | Hugging Face |
| Train Dataset | LLaVA-Pretrain | CC3M† + BSD-3-Clause | Hugging Face |
| Train Dataset | Open-LLaVA-NeXT-mix1M‡ | CC BY-NC 4.0 | Hugging Face |
| Evaluation Dataset | MME (MME-P) | arXiv Non-exclusive* | arXiv |
| Evaluation Dataset | VQAv2 | CC BY 4.0 | Official Website |
| Evaluation Dataset | MMBench | Apache 2.0 | GitHub |
| Evaluation Dataset | MMMU | Apache 2.0 | Hugging Face |
| Evaluation Dataset | MME (MME-C) | arXiv Non-exclusive* | arXiv |
| Evaluation Dataset | MathVista | CC BY-SA 4.0 | Hugging Face |
| Evaluation Dataset | MathVerse | MIT | Hugging Face |
| Evaluation Dataset | MathVision | MIT | Hugging Face |
| Evaluation Dataset | ScienceQA | CC BY-NC-SA | Official Website |
| Evaluation Dataset | AI2D | arXiv Non-exclusive* | arXiv |
| Evaluation Dataset | TextVQA | CC BY 4.0 | Official Website |
| Evaluation Dataset | DocVQA | arXiv Non-exclusive* | arXiv |
| Evaluation Dataset | ChartVQA | GPL-3.0 | GitHub |
| Evaluation Dataset | InfoVQA | arXiv Non-exclusive* | arXiv |

**Note †:** Subject to the CC-3M dataset may be freely used for any purpose, although acknowledgement of Google LLC ("Google") as the data source would be appreciated. The dataset is provided "AS IS" without any warranty, express or implied. Google disclaims all liability for any damages, direct or indirect, resulting from the use of the dataset

**Note ‡:** Our training data is based on Open-LLaVA-NeXT-mix1M, which combines ShareGPT4V data (CC BY-NC 4.0, research-only, with restrictions from LLaMA, Vicuna, and GPT-4 licenses) and AVG-LLaVA (Apache 2.0). All data was used strictly for academic research.

## G  LLM Usage

LLM is used to generate the math reasoning date for stage-3 training. See details in B.3.

## H  Concurrent Work

Concurrent to this work, MMaDa[78] (released after submission deadline) proposed a unified understanding and generation modeling using DM formulation. However, they do not leverage many of the novel techniques that we propose, leading to inferior performance and slow inference speed. We list a brief comparison in Table 14.

Table 14: **Comparison with MMaDa.** We report scores on MME, MMMU, and MMB benchmarks, along with average latency for image captioning.

| Model | MME | MMMU | MMB | Latency (s/image) |
|---|---|---|---|---|
| LaViDa-Dream | **1463.5** | 42.6 | **73.8** | **1.13** |
| LaViDa-LLaDa | 1365.6 | **43.3** | 70.5 | 1.32 |
| MMaDa | 1410.7 | 30.2 | 68.5 | 7.68 |

