# OpenReview forum: "LaViDa: A Large Diffusion Language Model for Multimodal Understanding"
_NeurIPS.cc/2025/Conference — NeurIPS 2025 spotlight_

### Official Review · Reviewer_fbuJ · 2025-06-19

**Clarity:** 3
**Significance:** 2
**Originality:** 3
**Rating:** 4
**Confidence:** 4

**Summary:**

To address the challenges of fast inference and controllable generation in vision-language models (VLMs), this paper proposes a novel paradigm based on discrete diffusion models (DLMs), leveraging their parallel decoding and bidirectional context-awareness. The method follows a LLaVA-style framework, where visual features are projected into the language embedding space via an MLP, and instruction tuning is employed to unlock the pretrained LLM’s multimodal capability. To tackle practical issues in training and inference, the authors introduce several techniques: complementary masking for effective training, prefix KV caching for efficient inference, and timestep shifting for high-quality sampling. Experimental results show the method achieves comparable or slightly better performance than some autoregressive VLMs, with additional benefits in speed and controllability.

**Questions:**

1. The paper repeatedly emphasizes the speed-quality trade-off, but the focus should be on quantitative comparisons with AR VLMs. Specifically, assuming comparable or better quality, how much faster is the DM model compared to SoTA AR-based VLMs?
- The COCO captioning task is not a good proxy for instruction-following benchmarks and may not fully reflect practical benefits.

2. What is the maximum context length supported by the model?
- If longer contexts (e.g., L = 4096) are required, would the speed advantage diminish significantly?
- The limited generation length is a serious bottleneck for real-world applications.

3. Bagel [ref1], under an AR framework, has shown parallel token output for image generation. Theoretically, it could also support parallel infilling of masked tokens—does this challenge the claimed advantages of the proposed approach?

4. Why not fully mask the answer segment? Was there any ablation comparing cases like "The [M] [M] dog." vs. "[M] answer is [M] [M]" vs. a fully masked " [M] [M] [M] [M] [M]"? What’s the impact of different masking strategies?

5. The model uses fewer visual embeddings compared to recent VLMs like Qwen2.5-VL. If the number of visual tokens is increased to match state-of-the-art models, would the inference speed advantage still hold?

6. InternVL-1.5 is fully open-sourced, and models like Mini-InternVL-Chat-2B/4B-V1.5 show significantly stronger overall performance than the proposed method.
- Could the performance improvements observed in this paper be primarily attributed to the visual encoder rather than the diffusion-based architecture itself?

[ref1] C. Deng, D. Zhu, K. Li, C. Gou, F. Li, Z. Wang, S. Zhong, W. Yu, X. Nie, Z. Song, G. Shi, and H. Fan. Emerging Properties in Unified Multimodal Pretraining. arXiv preprint arXiv:2505.14683, 2025.

**Ethical Concerns:**

["NO or VERY MINOR ethics concerns only"]

**Final Justification:**

- First VLM paradigm based on discrete diffusion models.
- Addresses practical challenges in integrating VLM tasks into diffusion language models. The Prefix-DLM effectively accelerates inference in multimodal settings.

**Limitations:**

yes

**Paper Formatting Concerns:**

NoA

**Quality:**

3

**Strengths And Weaknesses:**

**Strengths**
- First VLM paradigm based on discrete diffusion models.
- Addresses practical challenges in integrating VLM tasks into diffusion language models. The Prefix-DLM effectively accelerates inference in multimodal settings.
- The paper is clearly written and easy to follow.

**Weaknesses**
- The model is limited in generation length, constraining its applicability.
- While text-infilling is cited as an advantage over AR models, its practical usage scenarios are limited, making it a weaker justification.
- There remains a notable performance gap compared to leading open-source AR-based VLMs.

---

> ### Author Rebuttal · Authors · 2025-07-31
>
> **W1. Fixed-Length Generation Constraints of Diffusion Models**
>
> Overall, we recognize that fixed-length generation is a major constraint for diffusion language models. This limitation applies broadly across all such models. Notably, recent studies published after the submission deadline have begun to address this issue. For example, DreamOn [1] introduces special tokens that allow for expansion of masked tokens or deletion of unused ones. We will explore incorporating these advances in future work to address this limitation.
>
> **W2. Limited Use Cases for Text Infilling**
>
> We incorporate additional results on text-infilling across three tasks shown in Fig. 6: JSON satisfaction, text editing, and visually grounded script completion. We randomly sample 100 images from the LAION-2B dataset and use GPT-4o to generate task-relevant questions for each image. We also ask GPT-4o to produce optimized prompts for VLMs for these tasks. For the text editing and script completion tasks, the prompts instruct the VLMs to edit a specific sentence or complete a specific line while leaving the rest of the text unchanged. For JSON completion, we validate schema correctness using Python's json package. For text editing and script completion, we use string matching to verify whether the required prefix and suffix are preserved in the generated text. The results are reported in the table below.
>
> |  | JSON Satisfaction | Text Editing | Visually Grounded Script Completion |
> | --- | --- | --- | --- |
> | *LaViDa* | 1.0 | 1.0 | 1.0 |
> | *LaViDa-FIM* | 1.0 | 1.0 | 1.0 |
> | *LLaVa-1.6-7B* | 0.89 | 0.58 | 0.49 |
> | *Qwen2.5-VL-7B* | 0.98 | 0.85 | 0.22 |
>
>
> These results highlight the advantage of diffusion-based VLMs, which can provably achieve 100% constraint satisfaction without relying on careful prompt engineering. They also demonstrate the broader potential use cases for diffusion VLMs. We will incorporate more details of these evaluations in the final version.
>
> **W3. Performance Gap with SOTA**
>
> We acknowledge that our model currently shows a performance gap with state-of-the-art VLMs, particularly on OCR tasks. One contributing factor is that diffusion language models typically require more training steps to converge than autoregressive (AR) models, as shown in a recent study [2]. To test this in our setting, we conducted an additional two epochs of training, which resulted in substantial improvements on both OCR tasks (e.g., ChartQA) and conventional vision tasks (e.g., MME). These results suggest that additional training can significantly improve OCR performance. We hope that future work on diffusion language models will explore strategies for improving training efficiency. It is also worth noting that state-of-the-art VLMs are generally trained on more data and for more steps, which presents opportunities for further scaling.
>
> | Model                | MME   | MMBench | ChartQA | DocVQA | InfoVQA |
> | -------------------- | ----- | ------- | ------- | ------ | ------- |
> | Paper                | 341.1 | 70.5    | 64.6    | 59.0   | 34.2    |
> | +Additional Training | **444.3** | **75.6**   | **74.9**    | 68.6   | **43.4**    |
> | Open-Llava-Next-8B   | 336.8 | 74.4    | 69.7    | **69.9**   | 36.7    |
>
>
>
> **Q1. Additional Speed-Quality Tradeoff**
>
> As requested, we provide additional performance–latency results on MathVista in the table below. We report the maximum generation length (L), number of sampling steps, average number of non-padding tokens generated, per-sample latency, and throughput of non-padding tokens. Per review policy, we will include a visualization of these results similar to Figure 6 in the camera-ready version. To aid interpretation, we bold all metrics that outperform the AR baseline in the table. With optimal parameter choices, our model achieves up to a 2.5× speedup over the AR baseline.
>
> | Max Length | Steps | Avg Tokens | Acc(MathVista) $\uparrow$ | Latency(s) $\downarrow$| Throughput (tk/s) $\uparrow$|
> | --- | --- | --- | --- | --- | --- |
> | LaViDa-LLaDa |  |  |  |  |  |
> | 1024 | 256 | 378 | **45.5** | 22.0 | 17.2 |
> | 512 | 128 | 333 | **43.5** | **10.05** | **33.1** |
> | 256 | 256 | 220 | **45.2** | 16.44 | 13.4 |
> | 256 | 128 | 217 | **42.9** | **8.25** | **26.3** |
> | 256 | 80 | 215 | **44.9** | **6.25** | **34.4** |
> | 256 | 60 | 211 | **44.6** | **5.17** | **40.8** |
> | 256 | 40 | 216 | **43.6** | **4.18** | **51.7** |
> | 256 | 20 | 221 | 33.2 | **3.42** | **64.6** |
> | Open-Llava-Next-8B |  |  |  |  |  |
> | NA | NA | 199.75 | 42.0 | 10.48 | 19.1 |
>
>
> **Q2. Context Length**
>
> The context length is 4096 tokens. After accounting for vision and prompt tokens (~1000), the maximum possible generation length is around 3000 tokens. However, over 95% of our training data has generation lengths under 1000, which suffices for most tasks. As shown in the Q1 table above, even when AR models are allowed unlimited generation length, they typically produce only ~200 tokens on complex math reasoning tasks. When we set L = 1024 for LaViDa, it generates around 400 non-padding tokens. In summary, the current generation length is sufficient for most tasks, and within this range, the diffusion model demonstrates a speed advantage over AR models. We acknowledge that long-context generation remains underexplored for diffusion language models and hope that future work will address this area.
>
> **Q3. Bagel**
>
> Bagel uses a continuous flow-matching objective with an MSE loss for image tokens, enabling parallel image generation. However, it uses autoregressive next-token prediction with a causal mask for language modeling (see Appendix Fig. 15 of their paper). Therefore, it does not support parallel language generation. We also note that it was posted on arXiv on May 20th, after our submission deadline.
>
> **Q4.Why Not Fully Mask the Answer Segment?**
>
> During inference, we decode a fully masked sequence into clean text tokens in a gradual manner. To support this process, it is important to train the model with varying mask rates. In practice, we sample a random mask rate between 0 and 1 during training. For mask rates less than 1, we must adopt a masking strategy that makes the best use of the training data. We already include an ablation study comparing complementary vs. random masking in Table 4a, with further discussion in Appendix B.4. We did not experiment with always masking the full sequence because that essentially reduces to learning a one-step generation model, which is extremely difficult.
>
> **Q5.If the number of visual tokens is increased to match state-of-the-art models, would the inference speed advantage still hold ?**
>
> We extend the speed comparisons in Table 3a and Figure 4b to include an older AR model (LLaVA-1.5-7B), which uses low-resolution inputs (and thus fewer visual tokens). Our results show that LaViDa remains faster than AR models, even though it uses more visual tokens.
>
> |  | Res | CIDEr | Latency(s) |
> | --- | --- | --- | --- |
> | LaViDa | 768 | **114.8** | **1.23**|
> | Open-LlavaNext-8B | 1008 | 111.8 | 1.71 |
> | LLaVa-1.5-7B | 336 | 104.9 | 1.56 |
>
> **Q6(a). Intern-VL Vision Encoder**
>
> Small models such as Mini-InternVL achieve superior performance by distilling knowledge from stronger models. For example, their 300M vision encoder is distilled from InternViT-6B, a significantly larger model. Therefore, this is not a fair comparison. We plan to explore distilled vision encoders in future work.
>
> **Q6(b). Could the performance improvements observed in this paper be primarily attributed to the visual encoder?**
>
> We include ablations of different vision encoders in Appendix C, Table 11. We selected SigLIP for the final model due to its strong overall performance, although it is not the best on math tasks. For example, LaViDa-CLIP slightly outperforms LaViDa-SigLIP on MathVerse in our small-scale ablations (20.05 vs. 19.67). However, the full-scale model using the SigLIP encoder outperforms the baseline model using a CLIP encoder (27.2 vs. 14.6 in Table 1), suggesting that performance improvements do not stem solely from the vision encoder. In fact, our LaViDa-CLIP model trained on small-scale data still outperforms the AR baseline (19.67 vs. 14.6) using less data.
>
>
> [1] Zirui Wu*, Lin Zheng*, Zhihui Xie, Jiacheng Ye, Jiahui Gao, Yansong Feng, Zhenguo Li, Victoria W., Guorui Zhou, Lingpeng Kong (2025). DreamOn: Diffusion Language Models for Code Infilling Beyond Fixed-Size Canvas.
>
> [2] Prabhudesai, M., Wu, M., Zadeh, A., Fragkiadaki, K., & Pathak, D. (2025). Diffusion Beats Autoregressive in Data-Constrained Settings. arXiv preprint arXiv:2507.15857.

---

> > ### Comment · Reviewer_fbuJ · 2025-08-04
> >
> > Thank you for the additional experiments and clarifications. I am willing to keep my score.

---

### Official Review · Reviewer_Y6e3 · 2025-06-25

**Clarity:** 3
**Significance:** 3
**Originality:** 2
**Rating:** 4
**Confidence:** 3

**Summary:**

This paper introduces LaViDa, the first family of VLMs based on diffusion models. The authors propose several novel training and inference techniques for DMs, including complementary masking, Prefix-DLM, and timestep shifting that improve the training efficiency, inference speed, and sample quality of LaViDa. LaViDa achieves competitive performance across a wide range of tasks compared to AR VLMs, while offering the unique benefits of DMs.

**Questions:**

1. For complementary masking, if we ignore the time cost for vision encoding, then it is actually doing 2 forward-backward pass during training. However, according to Table 4a, the training time is only increased from 8.2hr to 8.9hr under the same training steps. Why is the time not doubled?
2. Could the authors report the total training time (gpu hours) for different stages for LLaViDA to see whether reproducing LLaViDA is possible for the research community.
3. What is the exact value for L when evaluating on general tasks (tasks involving short responses other than mathverse and mathvision)? What are the average **actual** response length for these tasks and what is the performance if we reduce L to the smallest possible and see the performance? This is to see whether LLaViDA is trying to modifying its earlier response in later diffusion steps.

**Ethical Concerns:**

["NO or VERY MINOR ethics concerns only"]

**Final Justification:**

The authors addressed most of my concerns. However, the performance of the proposed diffusion MLLM still lags behind existing AR counterparts. This also mentioned by other revieweres. The authors are expected to scale up its performance in the future and include the additional experiments mentioned in the rebuttal in the final version. Nevertheless, we can still accept this paper due to its valuable exploration on diffusion MLLMs.

**Limitations:**

yes

**Quality:**

3

**Strengths And Weaknesses:**

## Strengths
1. This paper identifies the important efficiency challenges encountered when training a diffusion-based LLM, e.g., inefficiencies in loss computing and the incompatibility to KV-cache. Although these challenges are not restricted to diffusion MLLM, the authors propose solutions that leverage unique characteristics of MLLM.
2.LLaViDA achieves good quality-speed trade-off on certain tasks, demonstrating promising future for diffusion-based MLLM.
3. This paper is well-formulated and is easy to follow.

## Weakness
### Insufficient comparisons on speed-quality trade-off between AR-based MLLM and LLaViDA.
Specifically, the comparison in the paper (Table-3a) only include the captioning tasks on COCO with Open-LLaVA-Next 8B. However, to fully demonstrate the advantage of LLaViDA, the authors should report the speed-quality trade-off on other tasks that require long responses such as MathVerse and MathVision.

### Lack of details on the evaluation of text infilling tasks.
What are the datasets used and their statistics when evaluating the constrained poem task (Table 2)? Moreover, how to evaluate the Sentence and sample-level constraint satisfaction? What are the prompts used to evaluate these tasks? Is it possible that the prompts used during evaluation favours diffusion-based MLLM other than the AR-based counterpart?

---

> ### Author Rebuttal · Authors · 2025-07-31
>
> **W1 Additional Performance-Latency Results**
>
>
> As requested, we provide additional performance-latency results on MathVista in the table below. We report the maximum generation length (L), the number of sampling steps, the average number of non-padding tokens generated, the latency per sample, and the throughput of non-padding tokens. In accordance with review policy (no image allowed), we will include an additional visualization of these results similar to Figure 6 in the camera-ready version. For clarity, we highlight all metrics that outperform the AR baseline in **bold** in the table. With an optimal choice of parameters, our model achieves up to a 2.5x speedup compared to the AR baseline model.
>
>
> | Max Length | Steps | Avg Tokens | Acc(MathVista) $\uparrow$ | Latency(s) $\downarrow$| Throughput (tk/s) $\uparrow$|
> | --- | --- | --- | --- | --- | --- |
> | LaViDa-LLaDa |  |  |  |  |  |
> | 1024 | 256 | 378 | **45.5** | 22.0 | 17.2 |
> | 512 | 128 | 333 | **43.5** | **10.05** | **33.1** |
> | 256 | 256 | 220 | **45.2** | 16.44 | 13.4 |
> | 256 | 128 | 217 | **42.9** | **8.25** | **26.3** |
> | 256 | 80 | 215 | **44.9** | **6.25** | **34.4** |
> | 256 | 60 | 211 | **44.6** | **5.17** | **40.8** |
> | 256 | 40 | 216 | **43.6** | **4.18** | **51.7** |
> | 256 | 20 | 221 | 33.2 | **3.42** | **64.6** |
> | Open-Llava-Next-8B |  |  |  |  |  |
> | NA | NA | 199.75 | 42.0 | 10.48 | 19.1 |
>
>
> **W2 Evaluation of Text-Infilling Tasks**
>
> We create a constrained satisfaction test set that requires the model to generate four-line poems, where the first word of each line must begin with two specific letters. The first letter must be a consonant from the set {'b', 'c', 'd', 'f', 'g', 'h', 'n', 'p', 'r', 's', 't'}, and the second letter must be a vowel from the set {'a', 'e', 'i', 'o', 'u'}. The task asks the model to generate a poem with specific starting syllables (e.g., "ca") for each line. Sentence-level and sample-level satisfaction are computed by string matching to check if each line begins with the specified characters. Sample-level satisfaction is achieved if all lines satisfy the sentence-level constraints. The prompt template is shown exactly in Figure 4. We note that this is a reasonable prompt and that better prompting may further improve the performance of AR models. However, one of the key advantages of diffusion models is their ability to achieve 100% provable satisfaction of arbitrary constraints without the need for prompt tuning.
>
> To further validate this claim, we include additional results on text-infilling for three tasks shown in Figure 6: JSON satisfaction, text editing, and visually grounded script completion. We randomly sample 100 images from the LAION-2B dataset and ask GPT-4o to generate a relevant question of each category for each image. We also ask GPT-4o to generate optimized prompts for VLMs for these tasks. For text editing and script completion, the prompts specifically ask the VLMs to edit a specific sentence or complete a specific line while leaving the rest of the text unchanged. For JSON completion, we verify schema compliance using the Python json package. For text editing and script completion, we use string matching to verify that the prefix and suffix are preserved in the output. The results are reported in the table below.
>
> |  | JSON Satisfaction | Text Editing | Visually Grounded Script Completion |
> | --- | --- | --- | --- |
> | *LaViDa* | 1.0 | 1.0 | 1.0 |
> | *LaViDa-FIM* | 1.0 | 1.0 | 1.0 |
> | *LLaVa-1.6-7B* | 0.89 | 0.58 | 0.49 |
> | *Qwen2.5-VL-7B* | 0.98 | 0.85 | 0.22 |
>
> These results highlight the strength of diffusion-based VLMs, which consistently achieve a 100% constraint satisfaction rate without the need for careful prompting. We will provide additional details on these evaluations in the final version.
>
>
> **Q1 Complementary Masking**
>
> We observe that encoding and backpropagating through the vision encoder introduces considerable overhead. Since we split each image into 5 views, adding one more conversation effectively increases the batch size of the vision encoder by 5. We conducted additional measurements on 8 A100 GPUs. Using a per-GPU batch size of 2 with complementary masking results in 4.16 seconds per step, while a per-GPU batch size of 4 without complementary masking results in 7.01 seconds per step, highlighting the high overhead of vision encoding.
>
> Another contributing factor is that we pad each batch to the longest sequence. The expected maximum length of 4 randomly selected sequences is greater than that of 2 randomly selected sequences.
>
> **Q2 GPU Hours**
>
> Total training time is 902 A100 (80GB) hours for LaViDa-LLaDa and 1,768 A6000 (48GB) hours for LaViDa-Dream.
>
> **Q3 Generation Length**
>
> We already report the values of L in Appendix B, Table 6. We will provide detailed average generation lengths for each task in the final version. In general, the generation length for short-answer tasks is typically fewer than 10 tokens. Changing the maximum length does not affect performance or the number of generated tokens, as the prompt often includes format instructions such as "answer with a few words." However, when evaluating our reasoning model on MathVista, we found that maintaining a generation length greater than 200 tokens generally yields stable performance, whereas aggressively reducing it leads to performance degradation.
>
> As an example, we include the results of varying generation length for AI2D and MathVista in the tables below.
>
> | **Max Length** | **Avg Tokens** | **Acc(Ai2d)** |
> | --- | --- | --- |
> | 16 | 2 | **70.0** |
> | 4 | 2 | 69.9 |
>
> | **Max Length** | **Avg Tokens** | **Acc(MathVista)** |
> | --- | --- | --- |
> | 1024 | 378 | **45.5** |
> | 512 | 333 | 43.5 |
> | 256 | 220 | 45.2 |
> | 128 | 102 | 41.2 |
>
> [1] Prabhudesai, M., Wu, M., Zadeh, A., Fragkiadaki, K., & Pathak, D. (2025). Diffusion Beats Autoregressive in Data-Constrained Settings. arXiv preprint arXiv:2507.15857.
>
> [2] Wu, Z., Zheng, L., Xie, Z., Ye, J., Gao, J., Feng, Y., Li, Z., Wu, V., Zhou, G., & Kong, L. (2025). DreamOn: Diffusion Language Models for Code Infilling Beyond a Fixed-Size Canvas.

---

> > ### Author Response · Authors · 2025-08-05
> >
> > Thank you again for your time in reviewing our paper. As the discussion phase nears its end, we’d like to check if our rebuttal has addressed your concerns. If there are any remaining points requiring clarification, we're happy to provide further details. If your concerns have been resolved, we would appreciate your consideration in updating the score to reflect the new results and discussion. We're also open to continuing the dialogue if needed.

---

> > ### Comment · Reviewer_Y6e3 · 2025-08-08
> >
> > Thank you for the additional experiments and clarifications. I am willing to keep my score.

---

> ### Author Response · Authors · 2025-08-06
>
> Dear Reviewer Y6e3
>
> This is a friendly reminder that the end of discussion period is fast approaching. We have conducted several additional experiments to address your concerns. We would appreciate it if you could let us know if those concerns have been successfully address.  If you have any last-minute questions, feel free to let us know as well.
>
> Best
>
> Authors

---

### Official Review · Reviewer_kZHX · 2025-07-02

**Clarity:** 3
**Significance:** 4
**Originality:** 3
**Rating:** 5
**Confidence:** 3

**Summary:**

The authors propose LaViDa, a family of diffusion-based vision-language models that integrate a vision encoder with a diffusion language model. The architecture consists of a pre-trained diffusion LM augmented with a visual encoder for image tokens embeddings and a projection layer to align vision features with the text model’s embedding space. The model is trained in two stages: a vision-language pretraining where only the projection is learned (to align visual features with the diffusion LM), and a supervised fine-tuning on multimodal instruction-following data where all components (vision encoder, projection, diffusion LM) are optimized together for tasks like image QA, captioning, etc. This two-stage setup is analogous to how AR VLMs (like LLaVA) are built by first aligning vision features and then fine-tuning on instructions. The authors address several challenges of the diffusion language models, introducing two-level complementary masking enabling the loss always be computed for the crucial answer tokens, and prefix caching for vision prefix that enables faster inference for long multimodal input.

**Questions:**

I have the following questions for the authors:
1. What is the memory footprint for the diffusion-based VLMs in comparison to the similar-size autoregressive models?
2. How do you plan to improve results on the OCR-heavy tasks?
3. Do you have any ideas on how to adapt your approach to all-to-all models that can not only read, but also to produce images?

**Ethical Concerns:**

["NO or VERY MINOR ethics concerns only"]

**Final Justification:**

I would keep my score (5) and recommend to accept this paper for the publication on NeurIPS. I see this paper as one of the former works on developing diffusion-based models in vision-language domain that is quite solid in the current format, and can motivate researchers to develop this topic more.

**Limitations:**

yes

**Quality:**

3

**Strengths And Weaknesses:**

The main strengths of the paper are as follows:

1. The authors introduced first Diffusion-Based VLM with the deep analysis and ablation study. Adapting standard adapter-based vision module integration to the diffusion-based LLMs is indeed clear and promising idea. The authors investigate different approaches on how to cope with the diffusion-based challenges, thus, providing the researcher good starting point to improve and enhance diffusion-based VLMs.
2. New technical enhancement for diffusion-based VLMs were introduced. The authors provide interesting modifications for the masking procedure that is important for the model to take signal from the most valuable output tokens (the answer tokens). Also, it is profitable to include the prefix-caching for the multimodal input that is usually the most token-intensive.
3. Extensive ablations with FIM objectives are valuable for further research in diffusion-based VLM.

The main weaknesses of the paper are as follows:

1. While the proposed models show comparable results with autoregressive models, we still see that the diffusion-based models provide worse results on text-intensive and OCR tasks. Thus, it is still a question, how to improve this part of the model's performance.
2. Fixed-length generation constraints of the diffusion. While the authors proposed modification for training using FIM objective, it is still a challenge, how not to train model to produce padding tokens in the case of the random-length generation.
3. The authors propose caching for the lengthy multimodal input, however, I couldn't find the memory footprint for the final pipeline, based on the memory-speed tradeoff, for now, it still seems that diffusion-based models could be less profitable in comparison to the autoregressive ones.

---

> ### Author Rebuttal · Authors · 2025-07-31
>
> We thank the reviewer for the constructive feedback.
>
> **W1 Performance on OCR Tasks**
>
> We acknowledge that our model’s performance on OCR tasks leaves room for improvement. One contributing factor is that diffusion language models generally require more training steps to converge than autoregressive (AR) models, as shown in a recent study [1]. To validate this hypothesis in our setting, we conducted an additional 2 epochs of training. This resulted in substantial improvements on both OCR tasks such as ChartQA and conventional vision tasks such as MME. Hence, OCR performance can be improved through additional training. We hope that future work on diffusion language models will further enhance training efficiency.
>
>
> | Model                | MME   | MMBench | ChartQA | DocVQA | InfoVQA |
> | -------------------- | ----- | ------- | ------- | ------ | ------- |
> | Paper                | 341.1 | 70.5    | 64.6    | 59.0   | 34.2    |
> | +Additional Training | **444.3** | **75.6**   | **74.9**    | 68.6   | **43.4**    |
> | Open-Llava-Next-8B   | 336.8 | 74.4    | 69.7    | **69.9**   | 36.7    |
>
>
>
> **W2 Fixed-Length Generation Constraints of Diffusion Models**
>
> We recognize that fixed-length generation is a significant constraint for diffusion language models. This limitation applies to all current diffusion-based language models. However, recent studies released after our submission deadline have begun to address this issue. For example, DreamOn [2] introduces special tokens that allow for expanding masked tokens or deleting unused ones. We plan to explore incorporating such advances in future work to overcome this limitation.
>
> **W3 Memory Footprint**
>
> We use prefix caching primarily to improve inference speed (see Table 3.a). On the A5000 GPU, we do not observe a significant difference in peak memory usage between the two settings. For example, memory usage is 22,811 MB with caching and 21,668 MB without caching.
>
> **Q1 Memory Footprint**
>
> Please refer to W3 above.
>
> **Q2 OCR Tasks**
>
> Please refer to W1 above.
>
> **Q3 Multi-Modality**
>
> As part of an ongoing follow-up project, we have integrated a VQ-tokenizer and additionally trained the model on text-to-image generation tasks. We achieved a GenEval score of 0.75, outperforming previous state-of-the art masked generative model Meissonic[3] (0.54) and a current work MMaDa[4] (0.63) . These are preliminary results and will be part of a separate future submission. Nonetheless, these results indicate that it is entirely feasible to extend our setup toward building a unified model for both understanding and generation using a single diffusion objective.
>
>
>
> References
>
> [1] Prabhudesai, M., Wu, M., Zadeh, A., Fragkiadaki, K., & Pathak, D. (2025). Diffusion Beats Autoregressive in Data-Constrained Settings. arXiv preprint arXiv:2507.15857.
>
> [2] Zirui Wu*, Lin Zheng*, Zhihui Xie, Jiacheng Ye, Jiahui Gao, Yansong Feng, Zhenguo Li, Victoria W., Guorui Zhou, Lingpeng Kong (2025). DreamOn: Diffusion Language Models for Code Infilling Beyond Fixed-Size Canvas.
>
> [3] Bai, J., Ye, T., Chow, W., Song, E., Chen, Q. G., Li, X., ... & Yan, S. (2024, January). Meissonic: Revitalizing masked generative transformers for efficient high-resolution text-to-image synthesis. In The Thirteenth International Conference on Learning Representations.
>
> [4] Yang, L., Tian, Y., Li, B., Zhang, X., Shen, K., Tong, Y., & Wang, M. (2025). Mmada: Multimodal large diffusion language models. arXiv preprint arXiv:2505.15809.

---

> > ### Author Response · Authors · 2025-08-05
> >
> > Thank you again for your time in reviewing our paper. As the discussion phase nears its end, we’d like to check if our rebuttal has addressed your concerns. If there are any remaining points requiring clarification, we're happy to provide further details. If your concerns have been resolved, we would appreciate your consideration in updating the score to reflect the new results and discussion. We're also open to continuing the dialogue if needed.

---

> > > ### Author Response · Authors · 2025-08-06
> > >
> > > Dear Reviewer kZHX
> > >
> > > This is a friendly reminder that the end of discussion period is fast approaching. We have conducted several additional experiments to address your concerns. We would appreciate it if you could let us know if those concerns have been successfully address.  If you have any last-minute questions, feel free to let us know as well.
> > >
> > > Best
> > >
> > > Authors

---

> ### Author Response · Authors · 2025-08-06
>
> Dear Reviewer kZHX
>
> We just noticed that while we included the memory footprint of LaViDa as requested in our original response, you also asked for a comparison with the memory cost of similar-sized auto-regressive models.  We apologize for this oversight and included the modified version of our response as below.
>
> **W3 Memory Footprint**
>
> We use prefix caching primarily to improve inference speed (see Table 3.a). On the A5000 GPU, we do not observe a significant difference in peak memory usage between the two settings. For example, memory usage is 22,811 MB with caching and 21,668 MB without caching. **This peak memory was comparable to similar-sized AR model. Specifically, the AR model Open-Llava-Next-8B has a memory usage of 20,112 MB. Both results are obtained with batch size 1 on MathVista benchmark with a max generation length of 256, using the Flash-Attention-2 kernel.**

---

> > ### Comment · Reviewer_kZHX · 2025-08-07
> > **Official Comment by the reviewer kZHX**
> >
> > Thank you for such a detailed and insightful response, and for conducting additional experiments that improve the solidity of your work. It is an interesting insight that diffusion-based models require more compute to obtain favorable results. To further improve the work, I would suggest building scaling laws for various sizes of the model, starting from the smaller ones, to be able to understand the specificity of the training procedure of diffusion-based VLMs.

---

> > > ### Author Response · Authors · 2025-08-08
> > >
> > > We are glad to know that you find our responses insightful. With regard to modeling scaling laws, we agree that such experiments would be meaningful. However, this is not practical at this time because there is no open-sourced large language model that is not in the range of ~10B parameters. Specifically, LLaDa only comes with a 8B variant and Dream only comes with a 7B variant. Since we need a base diffusion language model to build a VLM, we cannot build diffusion VLM at other sizes.
> > >
> > > We note that [1] , which was Arxived recently on Jul 21, 2025, experimented with a family diffusion language models at different sizes. However, the model and code was not released as of today Aug, 7. We will include scaling results in the camera-ready version if these models become publicly available by then.
> > >
> > > While we cannot experiment with scaling laws of model sizes, we plan include discussions of the scaling law of data and training compute. Notably, we find that different tasks converge differently. For example, We note that our model trained on 25% of data (used in ablation study in Appendix C, Table 11) already outperforms the AR baseline (19.67 vs. 14.6) on MathVerse despite using less data. When trained with full dataset, our model significantly outperform AR baselines on math and reasoning tasks (Table 1). By contrast, as also noted by the reviewer, OCR tasks takes additional training to converege. Hence, the compute requirement of diffusion models varies from tasks to tasks, with some tasks (e.g. Math) converges faster  than others.
> > >
> > > We will include detailed plots of the performance trends across different tasks at different training compute. Unfortunately, due to the rebuttal policy we cannot include additional media here at this time.
> > >
> > > [1] Prabhudesai, M., Wu, M., Zadeh, A., Fragkiadaki, K., & Pathak, D. (2025). Diffusion Beats Autoregressive in Data-Constrained Settings. arXiv preprint arXiv:2507.15857.

---

### Official Review · Reviewer_tRob · 2025-07-03

**Clarity:** 3
**Significance:** 3
**Originality:** 3
**Rating:** 4
**Confidence:** 4

**Summary:**

This paper proposes a multimodal large diffusion model called LaViDa. It introduces complementary masking, Prefix-DLM cache and timestep shifting to improve efficiency and performance. Experiments show that LaViDa achieves comparable performance with autoregressive models, with higher speed-quality tradeoffs and controllability.

**Questions:**

1.Though not required, it is highly suggested to include discussions on concurrent works like LLaDa-V[1], MMaDA[2] and Dimple[3].
2.Typos:
a.Line 227: InternVL-38B -> InternVL3-8B
b.Appendix Line 618: "than, K depending" -> "than K, depending"
c.Appendix Table 5: compositio -> composition
d.Appendix Line 671: LaViDaon -> LaViDa on
e.Appendix Line 713: 8 -> Table 8
f.Appendix Line 741: Figure 7 -> Figure 9
[1] You, Zebin, et al. "Llada-v: Large language diffusion models with visual instruction tuning." arXiv preprint arXiv:2505.16933 (2025).
[2] Yang, Ling, et al. "Mmada: Multimodal large diffusion language models." arXiv preprint arXiv:2505.15809 (2025).
[3] Yu, Runpeng, Xinyin Ma, and Xinchao Wang. "Dimple: Discrete diffusion multimodal large language model with parallel decoding." arXiv preprint arXiv:2505.16990 (2025).

**Ethical Concerns:**

["NO or VERY MINOR ethics concerns only"]

**Final Justification:**

After carefully reading comments from all reviewers and authors' rebuttal, my most concerns are well addressed. Therefore, I am glad to maintain my postive score.

**Limitations:**

yes

**Quality:**

3

**Strengths And Weaknesses:**

Strength
1.The three proposed techniques help to address the issues of extending discrete diffusion models to multimodal tasks.
2.The authors provide extensive experiments and ablations on the design choices to validate the effectiveness of each component.
3.The writing and figures are clear and easy to follow.

Weaknesses
1.To examine the speed-quality tradeoff more thoroughly apart from Figure 4b, the authors should provide performance-latency results on more benchmarks.
2.Results on OCRBench should be included to further validate the limitation on high-resolution understanding.
3.Since text infilling serves as an evidence to the advantage of diffusion-based models, more **quantitative experiments** should be conducted, e.g. Compare the metrics against autoregressive methods on the tasks in appendix Figure 6.

---

> ### Author Rebuttal · Authors · 2025-07-31
>
> We thank the reviewer for the constructive feedback.
>
> **W1 Additional Performance-Latency Results**
>
> As requested, we provide additional performance-latency results on MathVista in the table below. We report the maximum generation length (L), the number of sampling steps, the average number of non-padding tokens generated, the latency per sample, and the throughput of non-padding tokens. In accordance with review policy (no image allowed), we will include an additional visualization of these results similar to Figure 6 in the camera-ready version. For clarity, we highlight all metrics that outperform the AR baseline in **bold** in the table. With an optimal choice of parameters, our model achieves up to a 2.5x speedup compared to the AR baseline model.
>
>
> | Max Length | Steps | Avg Tokens | Acc(MathVista) $\uparrow$ | Latency(s) $\downarrow$| Throughput (tk/s) $\uparrow$|
> | --- | --- | --- | --- | --- | --- |
> | LaViDa-LLaDa |  |  |  |  |  |
> | 1024 | 256 | 378 | **45.5** | 22.0 | 17.2 |
> | 512 | 128 | 333 | **43.5** | **10.05** | **33.1** |
> | 256 | 256 | 220 | **45.2** | 16.44 | 13.4 |
> | 256 | 128 | 217 | **42.9** | **8.25** | **26.3** |
> | 256 | 80 | 215 | **44.9** | **6.25** | **34.4** |
> | 256 | 60 | 211 | **44.6** | **5.17** | **40.8** |
> | 256 | 40 | 216 | **43.6** | **4.18** | **51.7** |
> | 256 | 20 | 221 | 33.2 | **3.42** | **64.6** |
> | Open-Llava-Next-8B |  |  |  |  |  |
> | NA | NA | 199.75 | 42.0 | 10.48 | 19.1 |
>
> **W2 OCR Performance**
>
> As requested, we provide additional results on OCR-Bench in the table below. We acknowledge that our model’s performance on OCR tasks has room for improvement. One contributing factor is that diffusion language models typically require more training steps to converge than AR models, as demonstrated in a recent study [1]. To validate this hypothesis in our setting, we conducted an additional two epochs of training. This resulted in substantial improvements on both OCR tasks such as ChartQA and conventional vision tasks such as MME, confirming that OCR performance can be enhanced with additional training. We hope future research on diffusion language models will further improve their training efficiency.
>
> | Model                | MME   | MMBench | ChartQA | DocVQA | InfoVQA | OCR Bench |
> | -------------------- | ----- | ------- | ------- | ------ | ------- | --------- |
> | Paper                | 341.1 | 70.5    | 64.6    | 59.0   | 34.2    | 404       |
> | +Additional Training | **444.3** | **75.6**   | **74.9**    | 68.6   | **43.4**    | 485       |
> | Open-Llava-Next-8B   | 336.8 | 74.4    | 69.7    | **69.9**   | 36.7    | **605**       |
>
>
> **W3 Additional Quantitative Experiments on Text-Infilling**
>
> As requested, we have included additional results on text-infilling for three tasks presented in Figure 6: JSON satisfaction, text editing, and visually grounded script completion. We randomly sampled 100 images from the LAION-2B dataset and employed GPT-4o to generate relevant questions for each image category. We also utilized GPT-4o to create optimized prompts for these VLM tasks. For the text editing and script completion tasks, the prompts explicitly instruct the VLMs to edit a specific sentence or complete a specific line while preserving the remaining text. For JSON completion, we verified the schema compliance using the Python JSON package. For text editing and script completion tasks, we employed string matching to ensure that the prefix and suffix were preserved in the generated output. The results are reported in the table below.
>
> |  | JSON Satisfaction | Text Editing | Visually Grounded Script Completion |
> | --- | --- | --- | --- |
> | *LaViDa* | 1.0 | 1.0 | 1.0 |
> | *LaViDa-FIM* | 1.0 | 1.0 | 1.0 |
> | *LLaVa-1.6-7B* | 0.89 | 0.58 | 0.49 |
> | *Qwen2.5-VL-7B* | 0.98 | 0.85 | 0.22 |
>
> These results highlight the strength of diffusion-based VLMs, which consistently achieve a 100% constraint satisfaction rate without the need for careful prompting. We will provide additional details on these evaluations in the final version.
>
> **Q1 Concurrent Works Discussion**
>
> We are aware of the concurrent works mentioned, which were released after our submission deadline. We will include relevant discussions and citations for these works in the revised version.
>
> **Q2 Typographical and Grammatical Errors**
>
> We thank the reviewer for identifying these editorial errors and will correct them in the final version.
>
> [1] Prabhudesai, M., Wu, M., Zadeh, A., Fragkiadaki, K., & Pathak, D. (2025). Diffusion Beats Autoregressive in Data-Constrained Settings. arXiv preprint arXiv:2507.15857.

---

> > ### Author Response · Authors · 2025-08-05
> >
> > Thank you again for your time in reviewing our paper. As the discussion phase nears its end, we’d like to check if our rebuttal has addressed your concerns. If there are any remaining points requiring clarification, we're happy to provide further details. If your concerns have been resolved, we would appreciate your consideration in updating the score to reflect the new results and discussion. We're also open to continuing the dialogue if needed.

---

> ### Author Response · Authors · 2025-08-06
>
> Dear Reviewer tRob
>
> This is a friendly reminder that the end of discussion period is fast approaching. We have conducted several additional experiments to address your concerns. We would appreciate it if you could let us know if those concerns have been successfully address.  If you have any last-minute questions, feel free to let us know as well.
>
> Best
>
> Authors

---

### Decision · Program_Chairs · 2025-09-17

**Decision:**

Accept (spotlight)

**Comment:**

The final ratings for this paper are unanimously positive  (one "Accept" and three "Borderline Accep").

The AC agrees with the reviewers' evaluation. This paper introduces LaViDa, a new exploration of vision-language models (VLMs) built upon discrete diffusion models. Reviewers acknowledged the originality of this work as one of the first to adapt diffusion models for multimodal instruction following.  Initial reviews shared several key concerns. The primary concerns included: 1) The performance gap with state-of-the-art AR models, particularly on OCR-related tasks. 2) The need for more comprehensive quantitative analysis of the claimed speed-quality tradeoff. 3) A lack of quantitative evidence for the practical advantages of text-infilling and controllability.

The authors provided a rebuttal that effectively addressed these points. They presented new experiments, including detailed latency benchmarks demonstrating a speedup over standard AR models, additional training runs that narrowed the OCR performance gap, and new quantitative results that verified the advantages of their model in constrained generation tasks.

Following the rebuttal, all reviewers confirmed that their major concerns had been resolved and maintained their positive ratings. While a performance gap with state-of-the-art AR models remains, the paper's value lies in its timely exploration of a new diffusion-based VLM architecture. Therefore, the AC recommends this paper for acceptance.